# Two-step voltage-sensor activation of the human $K_V7.4$ channel and effect of a deafness-associated mutation

Mario Nappi [1], Damon J. A. Frampton[2], Ali S. Kusay[2], Kaiqian Wang [2], S. Suheda Yasarbas [2], Serena Pozzi[2], Francesco Miceli[1], Sara I. Liin [2], Maurizio Taglialatela [1] ✉ & Antonios Pantazis [2,3] ✉

*KCNQ4*-encoded $K_V7.4$ voltage-gated potassium channels are expressed in hair-cells of the inner ear. Loss-of-function variants in *KCNQ4* cause non-syndromic progressive hearing loss (DFNA2). $K_V7.4$ pore opening requires voltage-dependent conformational changes (activation) of the voltage-sensor domains (VSDs); however, how fast charge displacement during VSD activation is coupled to slow channel opening is currently unclear. Here, we optically tracked $K_V7.4$ VSD activation with voltage-clamp fluorometry, leveraging two fluorophores and pulsed excitation, and found that VSD activation comprises several voltage-dependent transitions, some with kinetics and voltage-dependence matching those of channel opening and closing. The DFNA2-causing R216H mutation impairs VSD movement and channel opening by destabilizing the active VSD configuration, a result confirmed by molecular dynamics simulations. We propose that the $K_V7.4$ VSD activates in two steps: a fast movement representing a first transition to an intermediate activation state, followed by slower component(s) that fully activate the VSD and drive channel opening.

The five genes of the *KCNQ* family (*KCNQ1-KCNQ5*) encode the subunits of voltage-gated potassium channels, designated $K_V7.1$ through $K_V7.5$. Among them, $K_V7.4$-channel subunits are expressed in inner and outer hair cells (IHCs and OHCs) of the cochlea and in central auditory pathways of the brainstem[1,2], as well as in skeletal muscle[3] and vascular smooth muscle cells[4]. Loss-of-function (LoF) mutations in *KCNQ4* cause non-syndromic autosomal dominant deafness (DFNA2)[1], a progressive form of sensorineural hearing loss.

Each $K_V7.4$ subunit comprises six transmembrane segments (S1-S6), with the voltage-sensor domain (VSD) formed by S1-S4, while S5 and S6 contribute to the central pore domain (Fig. 1a). These subunits assemble into functional channels, either as homotetrameric or heterotetrameric complexes with other $K_V7$-family members[5]. Channel opening occurs in response to membrane depolarization, whereupon

the positively charged S4 helix (Fig. 1b) moves outward to its activated state. This conformational change is then transmitted to the pore, a process referred-to as electromechanical coupling, opening the gate and permitting $K^+$ flow[6,7].

VSD activation has been characterized in some members of the $K_V7$ family[8–14]. The $K_V7.1$ VSD activates in two steps, first transitioning from resting to intermediate and then to the fully activated state. Both the intermediate and fully-activated states are coupled to distinct conducting states, with different biophysical and pharmacological properties[11,13,15,16]. In contrast to $K_V7.1$, the VSDs of $K_V7.2$ and $K_V7.3$ channels do not assume an intermediate state[14,17]. In several previous studies, disease-associated mutations in $K_V7$ channels have been utilized to inform on the gating mechanism of the channels and provide a molecular etiology[12,14,17–19]. Using gating-current measurements, we

[1]Department of Neuroscience, University of Naples "Federico II", Naples, Italy. [2]Division of Cell and Neurobiology, Department of Biomedical and Clinical Sciences, Linköping University, Linköping, Sweden. [3]Wallenberg Center for Molecular Medicine, Linköping University, Linköping, Sweden. ✉e-mail: maurizio.taglialatela@unina.it; antonios.pantazis@liu.se

previously reported that the $K_V7.4$ VSD activates faster and at potentials substantially more negative compared to channel opening[20]. These results raised a conundrum, of how fast and left-shifted VSD movements relate to the slow and depolarized $K_V7.4$-channel opening. Given the subtype-specific diversity in voltage-dependent gating among the $K_V7$ channels studied thus far, inferring VSD operation in $K_V7.4$ based on the activation mechanisms in $K_V7.1$-$K_V7.3$ is not possible.

In this work, to better characterize $K_V7.4$ VSD activation and its coupling to channel opening, we use voltage-clamp fluorometry (VCF) to optically track VSD movements in conducting $K_V7.4$ channels. We performed VCF experiments in the absence or presence of R216H: a LoF DFNA2-associated mutation[21] that removes a putative voltage-sensing charge in S4. Finally, we complement our studies with structural simulations.

## Results

### Implementing VCF in $K_V7.4$ channels

In most VCF studies of voltage-gated ion channels, an environment-sensitive, thiol-reactive fluorophore is conjugated to a substituted cysteine at the S3-S4 extracellular linker. When the charged S4 traverses the membrane electric field upon depolarization, the environment of the attached fluorophore will change, and so will its fluorescence emission. In this way, VSD activation can be tracked as ensemble fluorescence deflections ($\Delta F$)[22–26]. To select a labeling site appropriate for VCF recordings in $K_V7.4$, we engineered five cysteine mutants in the S3-S4 loop (Q194C, G195C, N196C, I197C and F198C, Fig. 1a, b) and used two-electrode voltage clamp to compare their function to that of the wild-type channel. We selected N196C for further study as it did not reduce current levels and only induced a modest shift in voltage dependence of approximately +10 mV (Fig. 1c, d, and Table S1). $K_V7.4$(N196C) channels are henceforth referred-to as $K_V7.4^*$.

Labeling $K_V7.4^*$ with Alexa Fluor 488 $C_5$ Maleimide (AF488) revealed robust $\Delta F$ (Fig. 1e). However, the prolonged excitation resulted in strong fluorescence photobleaching, which was challenging to effectively subtract for analysis. To limit the impact of photobleaching on the fluorescence signal we developed a pulsed excitation protocol for VCF, in which the excitation LED was on only 10% of the total time (100 μs every 1 ms). This maneuver eliminated most photobleaching, while leaving $\Delta F$ amplitude intact (Fig. 1f).

No $\Delta F$ were detectable in WT $K_V7.4$ labeled with AF488, confirming that the $\Delta F$ arose from fluorophores attached to C196, just outside S4 (Figure S1b). WT $K_V7.4$ labeled with AF488 exhibited a slightly hyperpolarized voltage dependence compared to unlabeled WT channels (Figure S1). Combined with the slightly depolarized opening of $K_V7.4^*$ (Figs. 1d, and S1), we later show that the voltage dependence of labeled $K_V7.4^*$ channel opening was practically wild-type-like.

### The $K_V7.4$ VSD activates during closed states and exhibits two kinetic components

Having identified an ideal labeling position and VCF protocol, we proceeded to characterize the VSD movements, as reported from $\Delta F$ signals, and relate them to channel opening and closing (Figs. 2, 3, and S2, S3). Voltage dependence was determined by fitting $\Delta F$ (VSD activation, or F(V)) and tail current (channel opening, or G(V)) to the Boltzmann distribution (Eq. 2 and 3, Fig. 2b). Interestingly, VSD activation had an apparent voltage-dependence shifted by about −20 mV relative to that of channel opening (Fig. 2b). A straightforward interpretation of this result is that the VSD undergoes conformational changes during transitions among closed states of the channel.

Kinetics were determined by fitting current or $\Delta F$ to single, or the sum of two, exponential distributions (Eq. 5; Fig. 3a). VSD activation

exhibited two kinetic components, one fast ($\tau_{fast} \cong 30–150$ ms) and one slow ($\tau_{slow} \cong 1$ s; Fig. 3a, c and S4a). The latter was subtle and exemplified why bleaching subtraction (here facilitated using pulsed excitation) is an important condition of good VCF practice. By contrast, a single exponential component did not provide a satisfactory fit to the $\Delta F$ (Fig. 3a). Channel opening was dominated by one component, which overlapped with the slow fluorescence component (Fig. 3b, c). Both VSD deactivation and channel closing exhibited two kinetic components, which overlapped with each other (Fig. 3d–f and S4b).

### The $K_V7.4$ VSD activates in two steps with distinct voltage dependence

The presence of two kinetic components for VSD activation and deactivation indicated that the voltage sensor undergoes distinct voltage-dependent transitions; however, this was not obvious in the apparent voltage-dependence of the AF488 $\Delta F$ signal (Fig. 2b). We hypothesized that an alternative fluorophore may report a composite VSD voltage-dependence more clearly. Accordingly, we repeated experiments in $K_V7.4^*$ using the 6-TAMRA C6 maleimide (TAMRA C6) fluorophore (Figs. 4, and S5, S6). In addition to different photochemistry (its excitation and emission wavelengths are red-shifted compared to AF488), the TAMRA C6 fluorophore possesses one more carbon link between the fluorescent and maleimide (cysteine-binding) moieties, presumably affording it greater length and flexibility[27].

TAMRA C6 $\Delta F$ (Fig. 4a) were qualitatively different from those of AF488 (Fig. 2a). Whereas AF488 $\Delta F$ increased consistently as the voltage became more positive, TAMRA C6 produced a more complex response to depolarization (voltage increase): progressive depolarizations up to about −30 mV resulted in fluorescence decrease. At −30 mV and above, an additional fluorescence component became apparent with opposite amplitude, which resulted in the $\Delta F$ increasing with progressive depolarizations (Fig. 4a, b). WT $K_V7.4$ showed no $\Delta F$ when labeled with TAMRA C6 (Figure S7), confirming that the composite $\Delta F$ arose from labeling C196. The two TAMRA C6 fluorescence components had distinct kinetics (as observed in AF488 $\Delta F$; Fig. 3); and the slower component had similar kinetics to channel opening (Fig. 4d, e and S4c).

We surmised that, since the slow $\Delta F$ component had similar kinetics to channel opening, it should also have similar voltage dependence. Accordingly, we fit the voltage dependence of TAMRA C6 $\Delta F$ to the sum of two Boltzmann distributions. The parameters of the first ($F_1$) component were free, while the second ($F_2$) component was constrained to be syntasic (i.e., have the same voltage-dependence[28]) with channel opening. This was achieved by fixing the $F_2$ voltage-dependence parameters (half-activation potential, $V_{0.5}$, and valence, $z$; Eq. 3) to those of the G(V) from the same cell. This resulted in an excellent fit to the data (Fig. 4f), with $F_1$ having a very negative voltage dependence ($V_{0.5} = -47$ mV). The two components had similar valence and $F_1$ accounted for 63% of the total $\Delta F$.

The TAMRA C6 fluorophore more clearly showed that the $K_V7.4$ VSD activates in two sequential steps, each with distinct voltage-dependence and kinetics. If the two fluorophores tracked the same VSD movements, why did the voltage-dependence of AF488 exhibit a single component? We hypothesized that AF488 also reported two voltage-dependent components, but they were difficult to discern. This hypothesis was supported by a re-evaluation of the AF488 voltage-dependence with the sum of two Boltzmann distributions, one of them being syntasic with that of channel opening (Fig. 4g). The two activation transitions of the $K_V7.4$ VSD were reported by both fluorophores, albeit differently.

### R216H impairs both VSD activation transitions and channel opening

How relevant are the $K_V7.4$ VSD transitions to channel opening? We sought to characterize the impact of a deafness-associated mutation,

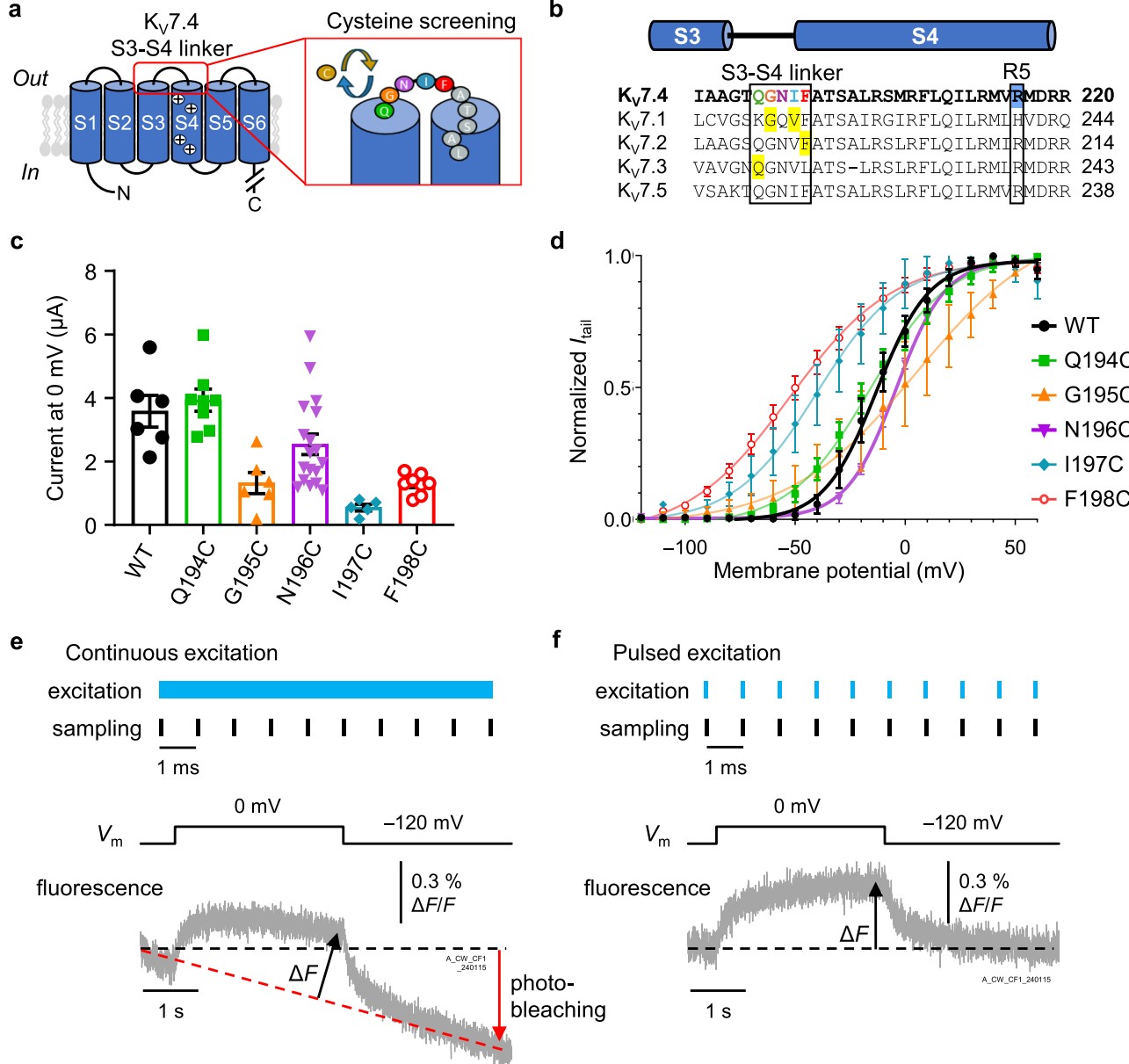

**Fig. 1 | Setting the stage for VCF experiments on $K_V7.4$. a** $K_V7.4$-subunit membrane topology and close-up on the S3-S4 linker. Four subunits assemble to form a functional channel. VSD = voltage-sensor domain; PD = pore domain. Cysteine scanning mutagenesis was performed for every residue in the linker. **b** Amino-acid sequence alignment for all human $K_V7$ channels, showing the S3-S4 linker and S4. Amino-acid positions labeled for VCF in studies of other $K_V7$ channels[8–10,14] are highlighted in yellow. R216 (R5) is highlighted in blue. **c** Current amplitudes for $K_V7.4$ WT (black; $n = 6$ cells) and S3-S4-linker Cys mutants Q194C (green; $n = 8$ cells); G195C (orange; $n = 6$ cells); N196C (purple; $n = 18$ cells); I197C (blue; $n = 5$ cells); F198C (red; $n = 8$ cells). **d** Mean, normalized voltage-dependence of channel opening in $K_V7.4$ WT and S3-S4-linker Cys mutants; colors and number of cells as in

panel c. Voltage-dependence parameters and statistical analysis in Table S1. The N196C substitution is thenceforth referred-to as $K_V7.4*$. **e** Gray trace shows fluorescence from $K_V7.4*$ labeled with AF488. The voltage pulse resulted in a fluorescence change ($\Delta F$, reporting VSD activation). In traditional VCF protocols, fluorescence excitation is constantly applied (blue bar). Over time this causes photobleaching (red dashed line), even between data sampling, while no emission is recorded. **f** As in panel e, using a pulsed VCF protocol. Here, the excitation LED was on 10% of the total time (100 μs every 1 ms), resulting in 90% less bleaching, as compared to the fluorescence trace from the same cell as in panel e. Error bars are SEM.

R216H[21], on VSD activation and channel opening, for two reasons: first, to provide a molecular etiology for the disease, and second, because the variant produces an amino-acid substitution at a VSD position likely important for its activation. R216 is a conserved, positively charged arginine in S4, homologous to the fifth arginine at the S4 of the archetypal Shaker voltage-gated $K^+$ channel (R5, Fig. 1b). We used this clinically relevant variant as a biophysical tool, to probe the link between VSD activation and channel opening.

VCF data from $K_V7.4*$ R216H channels (Fig. 5a–c, and S8–S12) were qualitatively similar to those from $K_V7.4*$ (Figs. 2a, 3d, 4a, S2, S3, S5, S6).

Our analysis revealed that R216H impaired the voltage dependence of channel opening and both VSD activation components (Fig. 5d–f). Specifically, channel opening and $F_2$ (which was constrained to have the same voltage-dependence) were shifted by 19 or 26 mV to more positive potentials, depending on whether the AF488 or TAMRA C6 label was used. $F_1$ was also affected, showing a 12-mV shift with both fluorophores. R216H also reduced the effective valence of these processes, by 7-34%. A straightforward interpretation of this analysis is that R216H hampered both VSD activation components and channel opening, such that they required more depolarization to occur (right-shifted $V_{0.5}$); and reduced

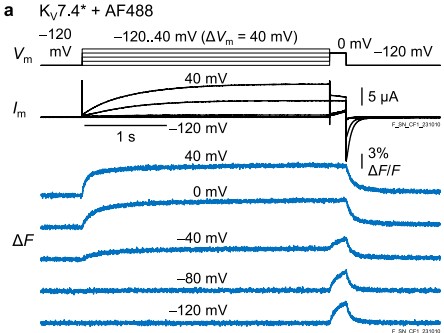
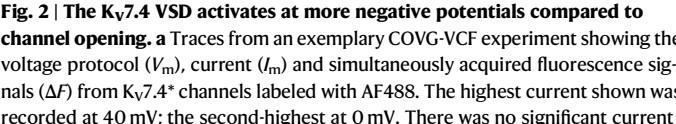
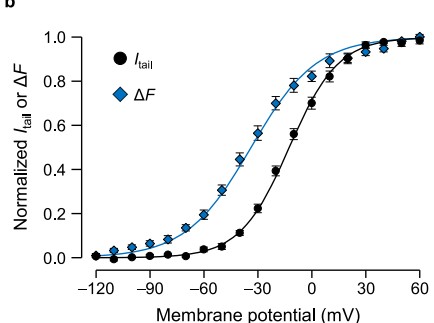

**Fig. 2 | The $K_V7.4$ VSD activates at more negative potentials compared to channel opening. a** Traces from an exemplary COVG-VCF experiment showing the voltage protocol ($V_m$), current ($I_m$) and simultaneously acquired fluorescence signals ($\Delta F$) from $K_V7.4^*$ channels labeled with AF488. The highest current shown was recorded at 40 mV; the second-highest at 0 mV. There was no significant current recorded at the more negative voltages, so the traces appear super-imposed. Traces at more potentials are shown in Figure S2. **b** Mean, normalized voltage-dependence of channel opening ($I_{tail}$, filled black circles) and VSD activation ($\Delta F$, filled blue diamonds) in $K_V7.4^*$ labeled with AF488, from 11 cells / 4 batches. Curves represent fits to Boltzmann distributions (Eq. 2 and 3, Table 1). Error bars are SEM.

their sensitivity to changes in voltage (reduction in effective valence). To confirm that both VSD activation transitions were inhibited by R216H, we attempted to fit the $\Delta F$ from $K_V7.4^*$ R216H channels constraining either $F_1$ or $F_2$ to have similar voltage-dependence parameters to those from $K_V7.4^*$ channels. This resulted in less satisfactory fits to the data (Figure S13), supporting that R216H impairs both transitions. Consistent with this analysis, we also report a shift by about 16 mV when AF488 $\Delta F$ data were fit to a single Boltzmann distribution, concomitant with a 30% valence reduction (Fig. 5f and S14a).

We also compared the activation and deactivation kinetics of $K_V7.4^*$ and the $K_V7.4^*$ R216H mutant (Figs. 5g-i, S4e-g, S14e,f). R216H apparently accelerated the kinetics of channel opening, (for the depolarization to 0 mV: $\tau_{R216H}/\tau_{WT} = 0.44 \pm 0.065$, $p = 0.023$). However, the $K_V7.4^*$ R216H current kinetic measurements were likely affected by inactivation, observed in this mutant (Fig. 5a, b), and low open probability[29]. Supporting this, there was no significant difference between the kinetics of the slow AF488 fluorescence component (which tracked well channel opening in the wild type): $\tau_{R216H}/\tau_{WT} = 0.84 \pm 0.12$, $p = 0.17$. The fast $\Delta F$ component of the AF488 activation signal was significantly accelerated $\tau_{R216H}/\tau_{WT} = 0.51 \pm 0.022$, $p = 2.6\text{E}{-}4$. We also observed an acceleration of the fast channel closing component (at the −80-mV repolarization; $\tau_{R216H}/\tau_{WT} = 0.69 \pm 0.033$, $p = 0.047$) concomitant with an acceleration of the fast AF488 $\Delta F$ signal ($\tau_{R216H}/\tau_{WT} = 0.42 \pm 0.018$, $p = 5.6\text{E}{-}6$). There was no significant difference in the slow components of channel closing and AF488 $\Delta F$ ($\tau_{R216H}/\tau_{WT} = 1.3 \pm 0.093$, $p = 0.13$ and $\tau_{R216H}/\tau_{WT} = 1.2 \pm 0.082$, $p = 0.066$, respectively).

TAMRA C6 $\Delta F$ kinetics were also generally accelerated (Fig. 5g; 0-mV depolarization: fast component: $\tau_{R216H}/\tau_{WT} = 0.44 \pm 0.050$, $p = 1.2\text{E}{-}4$; slow: $\tau_{R216H}/\tau_{WT} = 0.32 \pm 0.075$, $p = 0.0033$). We could not reliably fit the deactivation fluorescence with the sum of two exponentials (Figure S11), so it is not possible to make a straightforward comparison to the signal from wild-type channels. Likely, the opposite direction of the $\Delta F$ components, which effectively 'annihilate' each-other reduced the reliability of the kinetic measurements.

### R216H decreases S4 positional stability

The voltage-dependence shifts induced by R216H suggested that the mutation decreases the stability of the active VSD states relative to the resting. To assign a structural interpretation to this result, we conducted molecular dynamics (MD) simulations of $K_V7.4$ with the WT sequence and with the R216H mutant. Specifically, we used the apo $K_V7.4$ structure (PDB: 7BYL)[30] with S4 in a depolarized conformation, likely corresponding an activated VSD state.

First, the structural stability of WT or R216H channels in our simulation framework was assessed using root mean square deviation (RMSD) of residues in the transmembrane segments. The RMSD plots indicate that within the first ~200 ns of simulation, the proteins had an overall deviation of around 2.7 Å relative to the initial structures (Figure S15), which is consistent with thermal noise. Subsequent analyses were conducted on simulation frames from 200 ns and above to avoid frames where the protein was still equilibrating.

To examine how R216H affected VSD structural stability, we measured the distances between the Cα of R216 or H216 and the Cα of F143 (likely the charge-transfer-center residue[31] of $K_V7.4$) during our simulations (Fig. 6a). There was a wider distribution of distances in the R216H mutant compared to WT, and both distributions were centered around 12.5 Å (Fig. 6b). This indicates that the mutant position H216 fluctuated more relative to the charge transfer center than the WT R216. The same was true for the intracellular half of the S4 helix bearing either R216 or H216. We observed that RMSD was shifted further from the initial structures and had a wider distribution in the presence of the mutant (Fig. 6c). To determine whether the elevated motion of S4 containing the H216 mutant was primarily horizontal or vertical, we projected the Cα coordinates of H216/R216 and the reference residue F143 onto the XY and YZ planes (Figure S16). In the XY projection, the H216 mutant's S4 helix showed greater horizontal displacement away from F143 than in the WT system. The YZ projection revealed an arc-shaped motion, with the lower portion of the S4 helix swinging further from F143 in the H216 mutant. Notably, a small but distinct subset of frames positioned H216 below its WT counterpart, suggesting that the voltage sensor more readily moves downward in the mutant, consistent with its rightward shift in activation voltage.

To rationalize the difference in S4 positional stability, we analyzed the propensities of R216 or H216 to form stabilizing, intramolecular interactions with other residues. A spectrum of simulation frames illustrating salt-bridge interactions involving R216 and H216 is provided in Fig. 6d. In simulations with the mutant, H216 was assigned the doubly protonated, +1-charge tautomer, to enable salt-bridge interactions with neighboring acidic residues. R216 formed prominent salt-bridge interactions with E146 in S2 (80±11 % of simulation frames). By comparison, H216, which is also protonated, but shorter and more conformationally restricted than R216, interacted with E146 far less frequently (9±5 %). Conversely, both R216 and H216 interacted with D178 (S3) with similar frequency (76±11 and 67±10 % of simulation frames, respectively). The diminished interaction between H216 and E146 from the S2-helix resulted in the bottom of the S4-helix swinging away from S2 (Fig. 6e). Apart from E143 and D178, no other residues formed prominent interactions (i.e., >10 % of simulation frames) with R216 or H216. The lack of interaction of H216 with E146 meant that the mutation-bearing S4 was less stably anchored, indicating that the

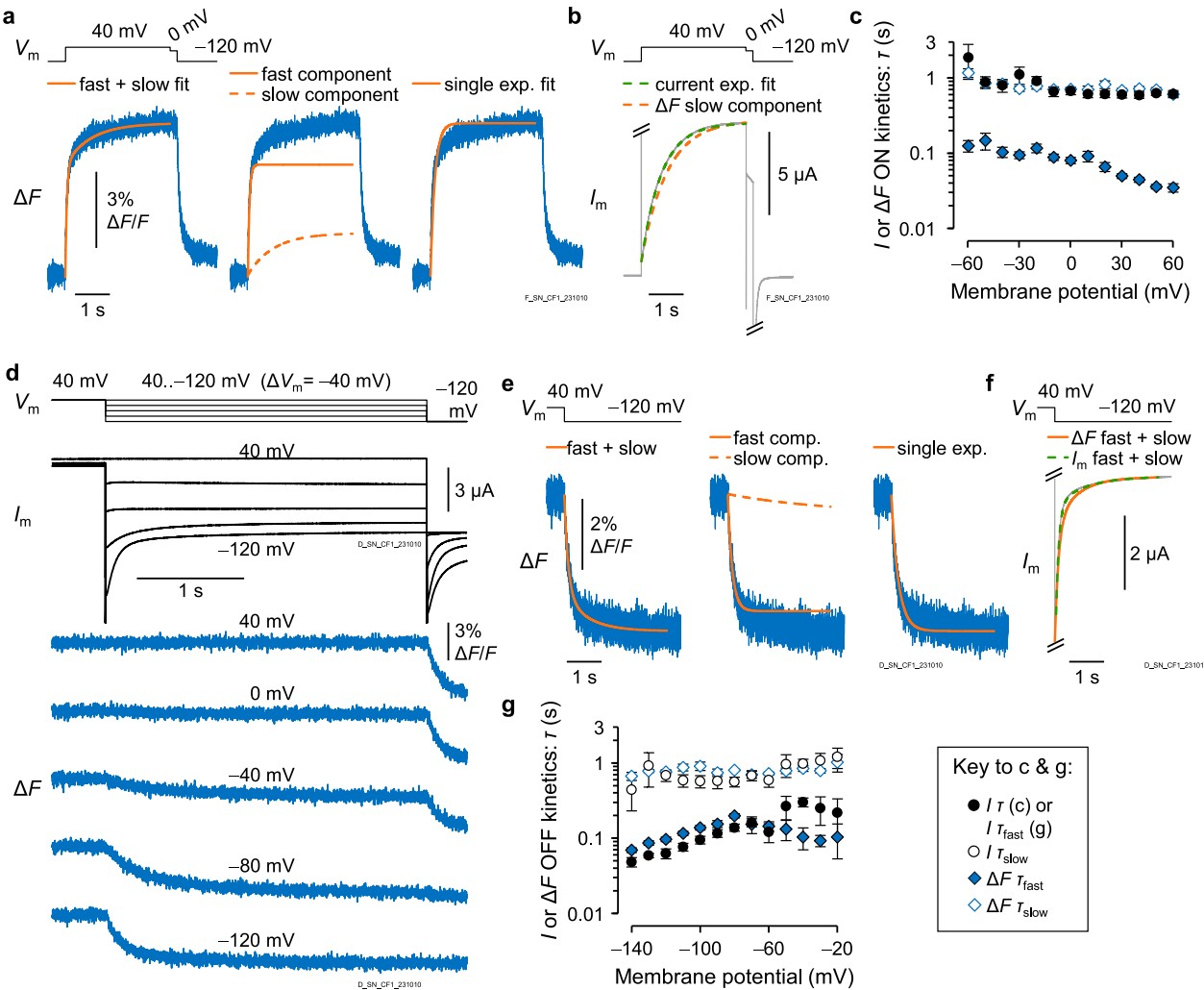

**Fig. 3 | $K_V7.4$ VSD activation and deactivation exhibit two kinetic components.**
**a** $\Delta F$ traces for the depolarization to 40 mV from Fig. 2a, fit to exponential functions
(Eq. 5, orange curves). The left trace was fit to the sum of two exponential functions.
The two components are shown individually in the center trace. Fast component
(solid): time constant ($\tau_{fast}$) = 0.044 s, 73 % fractional amplitude. Slow component
(dashed): $\tau_{slow}$ = 0.73 s. The right trace was fit to a single exponential function
($\tau$ = 0.13 s), which failed to capture the slow $\Delta F$ timecourse. Traces at more
potentials and their kinetic fits are in Figure S2. **b** Solid gray: The current trace for
the 40-mV depolarization from the same cell as in Fig. 2a and panel a. Dashed dark-
green: exponential function fit. Solid orange: the slow $\Delta F$ component (panel a,
center), scaled to the current amplitude. **c** Time constants ($\tau$) of channel opening
(filled black circles) and VSD activation ($\tau_{fast}$, filled blue diamonds; $\tau_{slow}$, open blue

diamonds) from 11 cells and 4 batches. $\tau_{slow}$ was similar to the rates of channel
opening. **d** Traces from an exemplary COVG-VCF closing/deactivation experiment
showing the voltage protocol ($V_m$), current ($I_m$) and simultaneously acquired
fluorescence signals ($\Delta F$) from $K_V7.4^*$ channels labeled with AF488. The highest
current was recorded at a 40-mV tail potential. **e, f** As in panels a and b, respectively,
for the deactivation pulse from 40 to −120 mV. Data from the same cell as in panel
d. The dashed dark-green curve is a kinetics fit. Traces at more potentials and their
kinetic fits are in Figure S3. **g** Time constants ($\tau$) of channel closing (fast: filled black
circles; slow: open black circles) and VSD deactivation (fast: filled blue diamonds;
slow: open blue diamonds) from 5 cells and 2 batches. The fractional amplitudes of
the kinetic components in panels c and g are in Figure S4a, b. Error bars are SEM.

activated VSD state in the R216H variant is less stable than in WT.
Notably, when H216 is in the neutral tautomeric form, all stabilizing
salt-bridge interactions would be abolished.

## Discussion

In this work, we sought to shed light on the voltage-sensing mechanism
of $K_V7.4$ channels. This involved five crucial steps: (1) identification of
N196, an ideal fluorescence labeling site on the S3-S4 extracellular
linker, which produced minimal functional effects on channel voltage-
dependent opening (Fig. 1a, b and S1); (2) pulsed excitation, which
largely abolished photobleaching (Fig. 1e, f); (3) use of two different
fluorophores, which helped resolve distinct VSD activation transitions
(Figs. 2–4); (4) use of the R216H deafness-associated mutation at an
eminent part of the VSD as a tool, to probe the relevance of VSD
transitions to channel opening (Fig. 5) and (5) leveraging a $K_V7.4$
structure to rationalize the effects of R216H with MD simulations

(Fig. 6). Thus we characterized the conformational changes under-
pinning the voltage-dependent development of physiological $K_V7.4$
currents, which contribute to, for instance, the repolarizing n-current
in OHCs and determining the vascular tone[32–35].

### The voltage-sensing mechanism of $K_V7.4$ channels

The data obtained in our optical investigation with both AF488 and
TAMRA C6 fluorophores revealed two movements of the $K_V7.4$ VSD,
with distinct kinetics and voltage dependence. The more parsimonious
interpretation of these data is that the same two VSD transitions are
reported (in terms of kinetics and voltage-dependence) by both
fluorophores. An alternative premise is that the $K_V7.4$ VSD undergoes
multiple transitions: one over very negative voltages ($V_{0.5} \cong$ −50 mV),
producing only the TAMRA C6 $F_1$ component (Fig. 4f, and Table 1);
another one or two, with intermediate voltage-dependence ($V_{0.5} \cong$
−30 mV; Fig. 2b, Table 1), giving rise only to the AF488 signals (Fig. 3a,

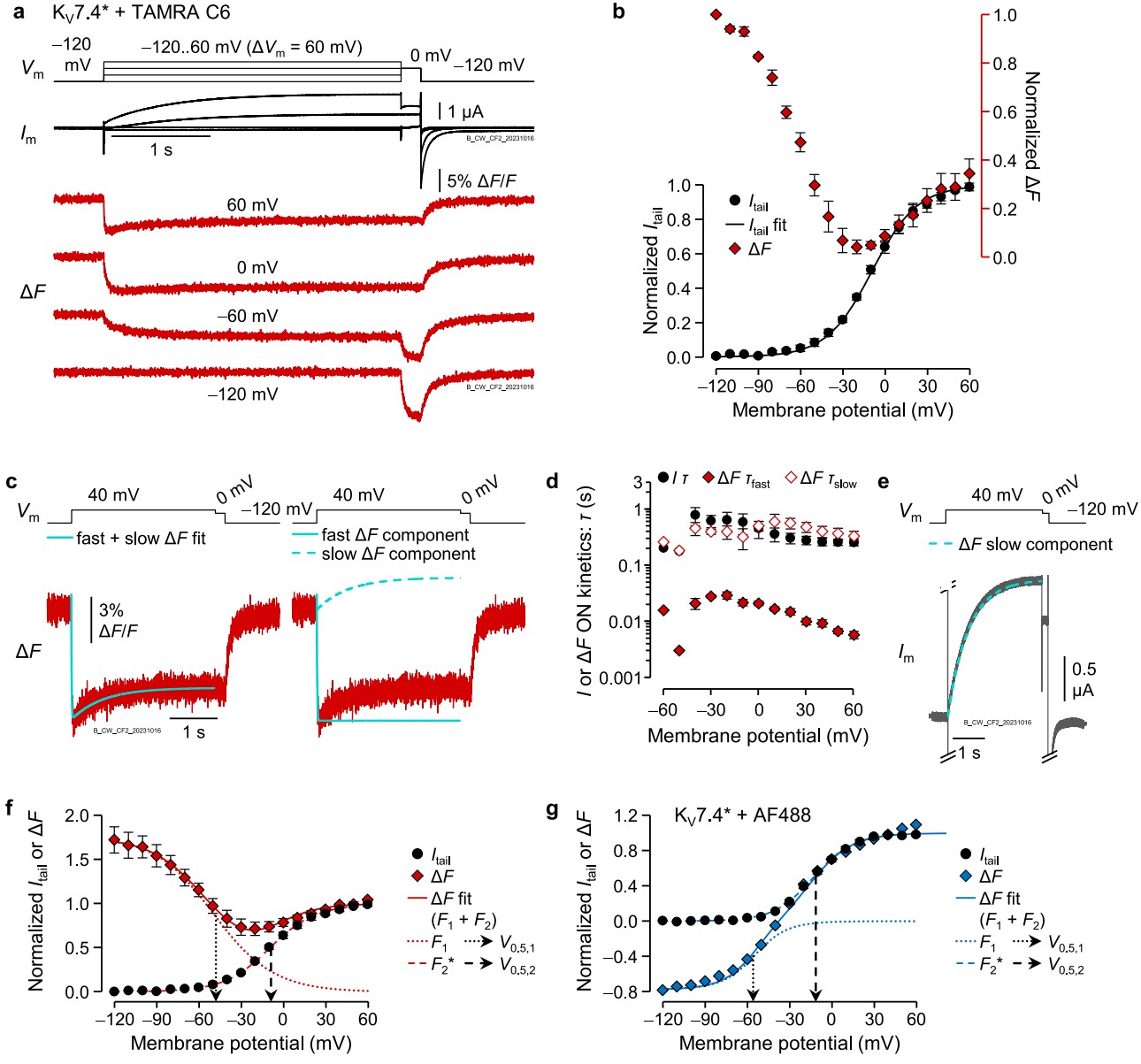

**Fig. 4 | A second VSD transition, resolved at voltages relevant to channel opening. a** An exemplary COVG-VCF experiment showing the voltage protocol ($V_m$), current ($I_m$) and simultaneously acquired $\Delta F$ from $K_V7.4^*$ channels labeled with TAMRA C6. The trace with highest current was at 60 mV, followed by 0 mV. **b** Mean, normalized channel opening ($I_{tail}$; black circles; left $y$-axis) fit to a Boltzmann distribution (black curve; Eq. 2, Table 1) and $\Delta F$ amplitude (normalized to minimal and maximal values; red diamonds; right $y$-axis). Data from 5 cells, 3 batches. **c** $\Delta F$ from panel. The left trace was fit to the sum of two exponential functions (Eq. 5; cyan). The two components are shown individually in the right. $\tau_{fast}$ = 7.3 ms, 78 % fractional amplitude (solid). $\tau_{slow}$ = 650 ms (dashed). Traces at more potentials and their kinetic fits, are in Figures S5–S6. **d** Mean time constants ($\tau$) of channel opening (filled black circles) and VSD activation ($\tau_{fast}$, filled red diamonds; $\tau_{slow}$, open red diamonds). Data from 5 cells, 3 batches. Fractional amplitudes are in

Figure S4c. **e** Current trace from the same cell as in panels a and c in gray. The dashed cyan curve is the slow $\Delta F$ component (panel c, right), scaled to the current amplitude. **f** Mean $\Delta F$ (red filled diamonds) fit to a double Boltzmann distribution (Eq. 3; solid red curve). The dotted red curve is the first component ($F_1$) and the dashed red curve is $F_2$ (filled black circles). The dotted arrow indicates the $F_1$ half-activation potential ($V_{0.5,1}$), and the dashed arrow shows $V_{0.5,2}$. Parameters in Table 1. The $\Delta F$ was normalized to the amplitude of $F_2$. Data from 5 cells, 3 batches. **g** AF488 $\Delta F$ (filled blue diamonds; from Fig. 2b) fit to a double Boltzmann distribution (Eq. 3). Blue curves are the $F_1$ component (dotted), $F_2$ (dashed) and their sum (solid). $\Delta F$ was normalized to the amplitude of $F_2$. $I_{tail}$ data are shown superimposed (filled black circles; from Fig. 2b). The dotted arrow indicates the $F_1$ half-activation potential ($V_{0.5,1}$), and the dashed arrow shows $V_{0.5,2}$. Parameters are in Table 1. Data from 11 cells, 4 batches. Error bars are SEM.

c); and a transition with a depolarized voltage-dependence matching that of pore opening ($V_{0.5} \cong 10$ mV), reported only as the $F_2$ component of TAMRA C6 (Fig. 4f, and Table 1). This premise requires that three or four distinct VSD transitions are each exclusively reported by the two fluorophores. This seems unlikely considering that the fluorophores label the same site and are similar in size.

It is not immediately clear how the two steady-state, voltage-dependence components ($F_1$, $F_2$) correlate with the two kinetic

components ($\tau_{fast}$, $\tau_{slow}$) observed during VSD activation and deactivation. Nevertheless, in channels labeled with TAMRA C6, both the depolarized ($F_2$) and the slow ($\tau_{slow}$) components match the voltage-dependence and kinetics of channel opening, respectively (Fig. 4d–f). We propose that $F_1$ represents the first activation transition of the $K_V7.4$ VSD, from a resting to an intermediate-active conformation; and $F_2$ a second transition, from the intermediate-active to a full-active state, which is linked to channel opening. This is illustrated in a scheme

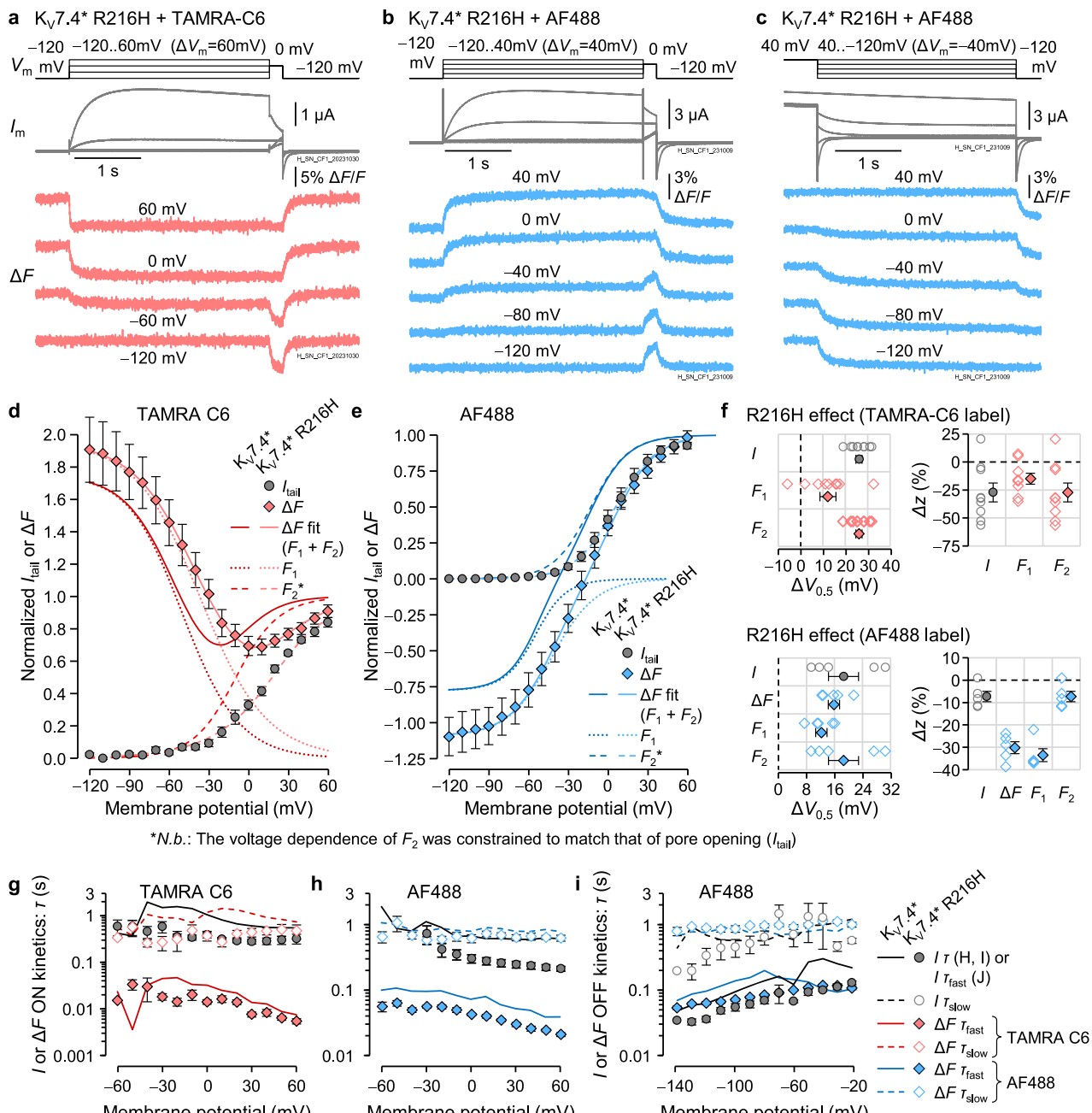

**Fig. 5 | R216H impairs both VSD activation transitions and channel opening.**
**a** An exemplary COVG-VCF experiment showing the voltage protocol ($V_m$), current ($I_m$) and simultaneously acquired $\Delta F$ from $K_V7.4^*$-R216H channels labeled with TAMRA C6. The trace with highest current was at 60 mV, followed by 0 mV. **b** As in panel a, from $K_V7.4^*$-R216H channels labeled with AF488. The current with highest current was at 40 mV, followed by 0 mV. **c** As in panel b, with a protocol for channel closing and VSD deactivation. The trace with highest current was at a 40-mV tail potential, followed by 0 mV and so on. **d** Mean, normalized voltage-dependence of channel opening ($I_{tail}$, gray circles) and VSD activation ($\Delta F$, light-red diamonds), from $K_V7.4^*$-R216H channels labeled with TAMRA C6. The $\Delta F$ was fit to a double Boltzmann distribution (Eq. 3, Table 1). Light-red curves are the $F_1$ component (dotted), $F_2$ (dashed) and their sum (solid). The $\Delta F$ was normalized to the amplitude of $F_2$. $\Delta F$ curves from $K_V7.4^*$ channels (dark red, from Fig. 4f) are also shown. **e** As in panel d, for $K_V7.4^*$-R216H channels labeled with AF488 ($\Delta F$ in light blue).

Parameters in Table 1. $\Delta F$ curves from $K_V7.4^*$ channels (dark blue, from Fig. 4g) are also shown **(f)** R216H effects: $V_{0.5}$ shifts ($\Delta V_{0.5}$) and percentile changes in $z$ ($\Delta z$), for channel opening (I, gray) and VSD activation (TAMRA C6: light red,; AF488: light blue). Open symbols: individual cells; filled symbols: means. Exact values and statistics are in Table S2. **g** Time constants ($\tau$) of channel opening (filled gray circles) and VSD activation ($\tau_{slow}$: filled light-red diamonds; $\tau_{fast}$: open light-red diamonds) for $K_V7.4^*$-R216H channels labeled with TAMRA C6. Lines show $K_V7.4^*$ data from Fig. 4d. **h** As in panel g, for $K_V7.4^*$-R216H channels labeled with AF488 (light blue is used for VSD kinetics). Lines show $K_V7.4^*$ data from Fig. 3c. **i** As in panel h, showing kinetics of channel closing and VSD deactivation. Lines show $K_V7.4^*$ data from Fig. 3g. Fractional amplitudes are in Figure S14b–d. All analysis on $K_V7.4^*$-R216H channels labeled with AF488 from 5 cells, 2 batches; TAMRA C6: 9 cells, 3 batches. Error bars are SEM.

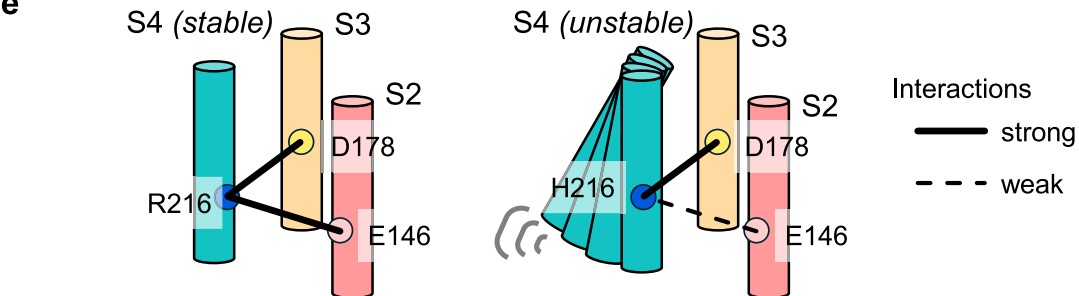

**Fig. 6 | R216H destabilizes the intracellular part of S4. a** Ensemble of simulation frames for $K_V7.4$ with WT sequence or the R216H mutant. 10 frames were uniformly captured over the last 300 ns of three simulation replicas (30 frames in total). The Cα atoms of F143, R216 and R216H are illustrated as orbs. Distances between Cα atoms of F143 and R216 or H216 are shown as dashed lines with more intense purple shades representing longer distances. The S1 helix is hidden for clarity. **b** Distance distribution for F143 – R216 and F143 – H216, measured between the residue Cα atoms. **c** Root mean squared deviation (RMSD) distribution of Cα atoms in the S4

helix residues (F208-D218) measured after aligning on the S1-S3 helices. **d** Alternative views of simulations frame ensembles of $K_V7.4$ to illustrate the side-chain conformations of E146, D178, and R216 or H216. The S1 helix is colored in light purple, the S2 helix in light red, the S3 helix in dark yellow and the S4 helix in teal. The simulations used the apo $K_V7.4$ structure (PDB: 7BYL)[30]. **e** Cartoon illustration of how the S4 helix is destabilized in the presence of H216 (right) compared the WT R216 (left) due to the weakening of its interaction with E146. A plot of F143 and R/H216 coordinates is shown in Figure S16.

in Fig. 7a. By extension, our interpretation for the relatively slow $K_V7.4$ opening is slow VSD activation, from the intermediate to the fully-active states (reported as $F_2$).

An alternative interpretation is that the two fluorescence components arise from distinct channel populations, such as conducting and non-conducting channels. For example, a structurally-locked conformation in which assembled channels are functionally dormant has been described for $K_V7.3$[36], although this seems to depend on the presence of an alanine residue at position 315, which is unique for $K_V7.3$

and is not conserved in $K_V7.4$. Thus, while we assume this is not the case here, we cannot definitively exclude this interpretation.

Another insight provided by studies of the R216H-mutant is the relevance of both VSD transitions to channel opening. The mutation results in the impairment (shift to more depolarized potentials) of both $F_1$ and $F_2$ VSD transitions, and channel opening (Fig. 5d–f and S14). MD simulations show that the mutation destabilizes an active VSD conformation, as H216 is less capable of forming stabilizing interactions with a counter-charge (represented by E146 in S2) than R216

**Table 1 | Voltage dependence parameters**

| Clone, fluorophore (Figure), measurement [n] | | $V_{0.5,1}$ (mV) | $z_1$ ($e_0$) | $f_1$ (%) | $V_{0.5,2}$ (mV) | $z_2$ ($e_0$) | b / o |
|---|---|---|---|---|---|---|---|
| $K_V7.4^*$ + AF488 (2b, 4g) | $I_{tail}$ | −13 ± 1.6 | 1.9 ± 0.085 | n/a | n/a | n/a | 4 / 11 |
| | $\Delta F$ [1] | −31 ± 1.8 | 1.2 ± 0.10 | 100 | n/a | n/a | |
| | $\Delta F$ [2] | −55 ± 1.3 | 2.3 ± 0.16 | 42 ± 3.8 | −13 ± 1.6 | 1.9 ± 0.085 | |
| $K_V7.4^*$ + TAMRA C6 (4f) | $I_{tail}$ | −9.1 ± 1.2 | 1.6 ± 0.14 | n/a | n/a | n/a | 3 / 5 |
| | $\Delta F$ [2] | −47 ± 5.8 | 1.3 ± 0.039 | 63 ± 2.0 | −9.1 ± 1.2 | 1.6 ± 0.14 | |
| $K_V7.4^*$(R216H) + TAMRA C6 (5d) | $I_{tail}$ | 17 ± 1.4 | 1.1 ± 0.13 | n/a | n/a | n/a | 3 / 9 |
| | $\Delta F$ [2] | −35 ± 3.6 | 1.1 ± 0.062 | 65 ± 2.3 | 17 ± 1.4 | 1.1 ± 0.13 | |
| $K_V7.4^*$(R216H) + AF488 (5e, S14a) | $I_{tail}$ | 6.1 ± 4.3 | 1.7 ± 0.043 | n/a | n/a | n/a | 2 / 5 |
| | $\Delta F$ [1] | −18 ± 1.6 | 0.98 ± 0.037 | 100 | n/a | n/a | |
| | $\Delta F$ [2] | −43 ± 1.6 | 1.6 ± 0.067 | 52 ± 2.8 | 6.1 ± 4.3 | 1.7 ± 0.043 | |

Parameters are from Eq. 2 ($I_{tail}$) and Eq. 3 & 4 ($\Delta F$). $n$ is the number of Boltzmann components (Eq. 3). $b$ and $o$ are the numbers of batches of oocytes and individual oocytes tested, respectively. Errors are S.E.M.

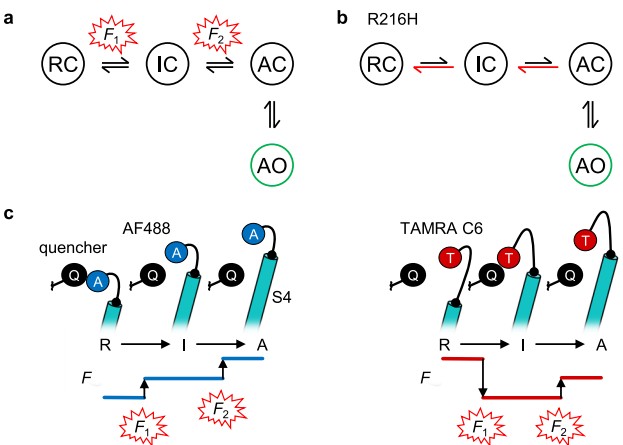

**Fig. 7 | Proposed mechanisms of $K_V7.4$ voltage-dependent activation, the effect of R216H, and VCF signals. a** In this scheme of channel states, the VSD can assume three conformational states: resting (R), intermediate (I) and active (A). The pore can assume two possible conformational states: closed (C) or open (O). We propose that only the fully-active state of the voltage sensor induces opening of the pore. The R ↔ I and I ↔ A transitions have distinct voltage-dependence and give rise to the $F_1$ and $F_2$ components in the fluorescence signals. **b** We propose that R216H destabilizes the intermediate- and fully-active VSD states, accelerating VSD deactivation transitions (red arrows), resulting in impaired VSD activation and channel opening. **c** Cartoon of a proposed mechanism for the differences in observed $\Delta F$ from AF488 and TAMRA C6 fluorophores. As S4 sequentially transitions from resting (R) to the intermediate (I) and active (A) states, it moves past a quencher (Q). AF488 (left) is quenched in the R-state; as S4 activates, it moves farther from the quencher, which reduces quenching probability and increases fluorescence emission. In this way, the macroscopic $\Delta F$ is monotonically increasing with voltage (Fig. 2). TAMRA C6 (right) is more flexible and experiences a different state-dependent interaction with the quencher, even while S4 undergoes the same movements as with AF488: the fluorophore is unquenched in the R-state; becomes more quenched in the I-state; and this quenching is partially relieved in the A-state. In this way, the macroscopic $\Delta F$ exhibits two macroscopic components with opposite amplitudes (Fig. 4).

(Fig. 6). This result is consistent with the accelerated VSD deactivation and pore closing observed in this mutant compared to wild-type (Fig. 5i). Destabilization of active conformations (Fig. 7b) accounts for the depolarizing shift in the steady-state voltage-dependence curves, and the loss-of-function phenotype, characteristic of deafness-associated variants of $KCNQ4$[1]. The MD simulations presented here reveal how the R216H mutation alters the local protein environment relative to WT, particularly affecting the dynamics of the lower S4 segment and the interactions formed by R216 versus H216. While local effects are evident, and the simulations are consistent with the

experimentally-observed destabilization of VSD activation, the R216H mutation likely influences protein dynamics beyond this region; longer or enhanced sampling simulations may provide insight into broader impacts, such as the downward movement of S4 during VSD deactivation.

## Optical and electrophysiological methods to measure $K_V7.4$ gating

The two VSD transitions were differentially reported by the two fluorescent probes. In the case of AF488, both VSD movements caused fluorescence unquenching (Figs. 2, 3a). In the case of TAMRA C6, $F_1$ resulted in fluorescence quenching and $F_2$ in unquenching (Fig. 4a, b, f). It has been previously reported that different fluorophores may experience different state-dependent interactions with nearby quenchers and report different $\Delta F$ signals[27,37–39]. Figure 7c shows a cartoon illustrating our hypothesis on how the same S4 movements can be differentially reported by the two fluorophores, given the same quenching mechanism.

A risk with any VCF investigation is that the fluorophores do not report on VSD conformational changes, but rather other voltage-dependent movements in the vicinity of the VSD. AF488 has been previously validated as a VSD-labeling fluorophore in several studies of voltage-gated channels, including the $K_V7$ family, ameliorating this concern[8–14,17,18,40–42]. TAMRA C6 is longer than AF488 by a single carbon linker (the length of a single C-C bond being 1.54 Å)[43]. The data it produces are compatible with an interpretation whereby it is quenched by the same mechanism as AF488 (Fig. 7c). In a previous VCF study on the BK channel, we used the TMR-5'/6'-C6-maleimide fluorophore, which consists of TAMRA C6 (as the 6' isomer) and its longer 5' isomer[27]. MD simulations showed that the fluorophore rarely stretches to its full length ( ~ 25 Å), spending most time folded within 17 Å of the label site[27]. Thus, although we acknowledge that the identity of the quenching mechanism is unknown, we do not anticipate that the two fluorophores report distinct VSD movements or voltage-dependent changes outside the VSD.

$F_1$ is most similar to the charge displacement reported in a gating-current study in $K_V7.4$[20], as both develop faster and at more negative potentials than channel opening. $F_2$ would have been challenging to detect with gating currents: First, because the amplitude of gating currents is proportional to the rate of charge movement, and $F_2$ develops with a rate three orders of magnitude lower than more typically resolved charge movements (seconds instead of milliseconds). Second, because charge integration over the time-scales necessary to capture $F_2$ would make measurements very susceptible to electrophysiological artifacts (noise, membrane instability). Third because, as $F_2$ exquisitely tracks with channel opening, the pore blockade necessary for gating current measurement may allosterically

alter $F_2$ properties (or blockers may directly perturb VSD movements). While VCF circumvents these shortcomings, $F_2$ was still challenging to detect optically, requiring confirmation with two fluorophores and a method to decrease fluorescence photobleaching over the particularly long recording times, explained below.

## Technical interlude: pulsed excitation to minimize photobleaching

Slow $K_V7.4$ kinetics necessitated the use of long recordings (up to 7 s), at which point the photobleaching matched or exceeded $\Delta F$ amplitude (Fig. 1e). To reduce photobleaching, we drew inspiration from a pulsed-excitation application in cardiac local-field fluorescence microscopy. In this work, Escobar and colleagues used high-power, picosecond laser excitation pulses (briefer than the mean fluorescence lifetime) to maximally excite rhod-2 fluorophores while minimizing secondary excitations and photobleaching[44]. In our experiments, the slow $K_V7.4$ kinetics allowed for a relatively low sampling rate (1 kHz). While our LEDs could not be pulsed at rates relevant to the fluorescence lifetime, it was possible to turn them on around the sampling point and keep them off for 90 % of the sampling interval, while data were not collected. In this way the $K_V7.4$-conjugated fluorophores received full excitation during sampling, producing robust $\Delta F$; while photobleaching was minimized, as the total light exposure was reduced by 90 % (Fig. 1e, f). This was straightforward to implement in a commonly used Digidata/Clampex acquisition system. Even more precise LED control can be achieved with specialized timers, as demonstrated in a pulsed-excitation VCF study of P2X7 channels[45].

## The $K_V7.4$ voltage-sensing mechanism in context

Considering other members of the $K_V7$-channel family, the $K_V7.4$ voltage-sensing mechanism recapitulates most closely that of $K_V7.1$ channels complexed with regulatory KCNE1 subunits[41]: the VSD undergoes two sequential activation transitions, and the second is thought to be coupled to opening. Similar to our findings for $K_V7.4$, only the second voltage-sensor movement overlaps in voltage with channel opening. Another similarity between $K_V7.4$ and $K_V7.1$/KCNE1 channels is the slow time-course of channel opening. In the absence of KCNE1, $K_V7.1$ channels can also open from the intermediate-active VSD state, do so much faster, and display F(V) and G(V) curves that largely overlap in voltage (although with noticeable $F_1$ and $F_2$ components). Largely overlapping F(V) and G(V) curves are reported also for $K_V7.2$ and $K_V7.3$ channels, although no stable intermediate-active state has been observed in these channels[14,17]. Altogether, $K_V7.4$ deviates from $K_V7.2$ and $K_V7.3$ by displaying a stable intermediate-active voltage sensor state, and deviates from $K_V7.1$ by not showing signs of ion conduction at this intermediate state.

Intriguingly, $K_V7.1$ channels have a histidine (H240) instead of the R5-arginine mutated in the $K_V7.4$ R216H variant (Fig. 1b). Substitution of H240 in $K_V7.1$ to arginine induced a leftward shift in the voltage dependence of the ionic and gating currents[46], mirroring the effects of R216H reported here. Moreover, the R216 (S4)/E146 (S2) interaction (Fig. 6) is observed also in the $K_V7.2$ VSD, involving homologous amino-acids R210 (Fig. 1b) and E140, and stabilizing the active VSD conformation[19,47]. Thus, the charge/countercharge interaction network appears to be conserved in at least two members of the $K_V7$-channel family.

In vivo, regulation of $K_V7.4$ channel gating has remarkable physiological implications. For example, in mammalian OHCs, $K_V7.4$ channels introduce a large conductance (also known as $IK_n$), active at the cell resting potential, which provides a critical contribution to their somatic length changes that amplify sound-evoked vibrations. Mature OHCs show clustering and intrachannel cooperativity of $K_V7.4$ channels[48], which is likely influenced by regulatory interactions with protein partners[5,49]. These include inhibition by $Ca^{2+}$-bound calmodulin, which interacts with cytosolic C-terminal helices[50], but also the S2-S3 loop in the VSD[51,52], and potentiation by the G-protein βγ complex[53]. We propose that VCF is the ideal tool to investigate such interactions *in cellula*, as in previous studies of $Ca^{2+}$- and Gβγ-dependent VSD regulation in $BK_{Ca}$ and $Ca_V2.2$ channels, respectively[54,55].

Finally, while our work here focused on the properties of $K_V7.4$ homo-tetramers, $K_V7.4$ subunits can also co-assemble with the KCNE4 auxiliary subunits in OHCs[56] and in vascular smooth muscle cells[57], where they also form heteromeric channels with other members of the $K_V7$ family, like $K_V7.5$[33,35]. It would be exciting in future studies to determine how such physiologically relevant interactions impact voltage sensor movement and channel opening.

# Methods

## Ethical statement

The use of animals, including the performed surgery, was reviewed and approved by the Linköping Animal Care and Use Committee of the Swedish Board of Agriculture (Case no. 1941 and 14515) and conforms to national and international guidelines (Directive 2010/63/EU of the European Parliament on the protection of animals used for scientific purposes).

## Oocyte preparation

Individual oocytes from *Xenopus laevis* frogs were acquired either through surgical removal followed by enzymatic digestion at Linköping University, or purchased from Ecocyte Bioscience (Dortmund, Germany). Oocytes at developmental stages V-VI were selected for experiments and injected as follows.

## Molecular biology

Human *KCNQ4* (GenBank accession no. NM_004700) inserted into a pXOOM vector was used in this study. cRNA for injection was prepared from DNA using a T7 mMachine transcription kit (Invitrogen, Stockholm, Sweden), and cRNA concentrations were determined spectrophotometrically (NanoDrop 2000c, Thermo Scientific, Stockholm, Sweden). Mutations were introduced through site-directed mutagenesis (QuikChange II XL, with 10 XL Gold cells; Agilent Technologies, Kista, Sweden) and confirmed by sequencing at the Molecular Biology unit of the Linköping University Core Facility. Mutagenesis primers were designed according to supplier instructions (Table S3). Constructs are available upon reasonable request.

## Two-electrode voltage clamp

The initial electrophysiological experiments to evaluate the voltage dependence and functional expression of Cys-substitution clones (Fig. 1c, d) were performed under two-electrode voltage clamp. Oocytes from developmental stages V-VI were injected with 50 ng of cRNA encoding either wild-type or mutant *KCNQ4* and incubated at 16 °C for 2-4 days prior to electrophysiological experiments. Two-electrode voltage clamp experiments were performed using a Dagan CA-1B amplifier (Dagan, MN, USA). Whole-cell $K^+$ currents were sampled using Clampex software (Molecular Devices, San Jose, CA, USA) at 5 kHz and filtered at 500 Hz. All experiments were carried out at room temperature (-20 °C). The extracellular recording solution contained (in mM): 88 NaCl, 1 KCl, 0.4 $CaCl_2$, 0.8 $MgCl_2$, and 15 HEPES. pH was set to 7.4 by addition of NaOH. Voltage-step protocols consisted of a − 80 mV holding potential and a hyperpolarizing −100 mV pre-pulse for 2 s to ensure channel closure. This was followed by test pulses ranging from −100 mV to +60 mV with incremental depolarizing steps of 10 mV for 3 seconds, to generate $K^+$ currents. The tail pulse was set to −30 mV for 2 s. Data were recorded under constant superfusion (rate set to 1.0 mL/s) with the extracellular recording solution.

## Cut-open oocyte Vaseline gap (COVG)

We used the cut-open oocyte Vaseline gap (COVG), a low-noise, fast voltage-clamp technique[58–60]. Each oocyte received 50 ng of $K_V7.4$

cRNA. Injected oocytes were incubated at 16 °C for 3-5 days prior to electrophysiological experiments. The oocyte was placed in a triple-compartment acrylic chamber, with a diameter of 600 μm for the top and bottom apertures. We used following solutions: external solution (mM): 120 Na-methanesulfonate (MES), 1 K-MES, 2 Ca(MES)$_2$, 10 HEPES (pH=7.0); internal solution (mM): 120 K-glutamate, 10 HEPES (pH=7.0); intracellular micro-pipette solution (mM): 2700 Na-MES, 10 NaCl. Low access resistance to the oocyte interior was obtained by permeabilizing the oocyte with 0.1% saponin carried by the internal solution. In order to evoke responses to measure G(V) relationship, each sweep was 5 s in duration with 10 s intersweep interval. The holding potential was −120 mV. Each sweep was composed of a 500-ms prepulse at the holding potential, followed by a three-second step increasing by 10 mV for each sweep from the holding potential. This was followed by a 0-mV step (200 ms), and finally a 1.5-s step at the holding potential. The 0-mV step was used to provide a measure of channel open probability at an isopotential point immediately after the test pulse. To measure deactivation kinetics, a prepulse consisting of holding the cell at −120 mV for 500 ms and then to 40 mV for 1 s was applied before every sweep. Acquisition then started and the cell was kept for another 500 ms at 40 mV, followed by a three-second step increasing by 10 mV for each sweep from the holding potential (−120 mV) to 0 mV. Every sweep was followed by 500 ms at the holding potential. The inter-sweep interval was 10 s. The current signal was filtered at 200 Hz and acquired at 1 kHz.

## Voltage-clamp fluorometry

Voltage-clamp fluorometry was performed by adding epifluorescence and light detection capability to the COVG system[60,61]. Prior to mounting on the COVG apparatus, oocytes were stained on ice for 30 min with 100 μM Alexa Fluor 488 C$_5$ Maleimide fluorophore (AF488; ThermoFisher) or for 5 minutes with 20 μM 6-TAMRA C6 maleimide fluorophore (TAMRA-C6; AAT Bioquest), in a depolarizing solution (in mM: 120 K-MeS, 2 Ca(MeS)$_2$, 10 HEPES, pH 7.0), on ice, in the dark, to label the introduced Cys (N196C) at the S4 extracellular flank. The oocytes were then rinsed in dye-free SOS prior to being mounted in the recording chamber. Fluorescence emission and ionic current were simultaneously measured from the same area of membrane isolated by the top chamber. The same electrophysiological apparatus and solutions were used as above, for COVG. For AF488 experiments, the optical set-up consisted of a BX51WI upright microscope (Olympus) with filters (Semrock BrightLine: exciter: FF01-482/35-25; dichroic: FF506-DI03-25×36; emitter: FF01-524/24-25). The excitation light source was a Thorlabs blue LED (490 nm, 205 mW, M490L4) driven by a Cyclops LED driver (Open Ephys). For TAMRA-C6 experiments, the following filter set was used: exciter: FF01-531/40-25; dichroic: FF562-Di02- 25×36; emitter: FF01-593/40-25 (all Semrock BrightLine). The light source was the M530L3 green LED (530 nm, 170 mW, Thorlabs). A LUMPLANFL 40XW water immersion objective (Olympus; numerical aperture = 0.8, working distance = 3.3 mm) and SM05PD3A Si photodiode (Thorlabs) were used for fluorescence detection. Photocurrent was amplified with a DLPCA-200 current amplifier (FEMTO). We can confirm that the pulsed excitation, described below, also worked with a Thorlabs DC2200 driver paired with the SOLIS-4C (lime green) LED. Tentative testing with the Thorlabs LEDD1B driver showed unsuitable timing (explained below).

We used pulsed excitation of the fluorophore to reduce photobleaching of the fluorescence signal during long-lasting protocols. Data were sampled at 10 kHz. The LED driver was triggered by a 5-V signal from a digitizer analog output. In Clampex, the LED trigger analog signal output was set to *Train*, with train period 10 samples and pulse width 1 sample. This resulted in a 1-kHz train rate with pulse width 100 μs (10% duty cycle). If the combination of the digitizer triggers and the LED driver and light were well timed, a 10-kHz record was acquired where one in every ten samples contained fluorescence

emission, and nine samples out of ten were dark, acquired in the complete absence of excitation. A combination with unsuitable timing resulted in the acquired emission being split over two or more samples. After acquisition, data were decimated 10-fold in MATLAB after first time-shifting the data such that the traces started with a light sample (this operation could also be performed in Clampfit). This resulted in a 1-kHz trace including only light samples, with similar fluorescence level as when emission was on constantly, yet with a fraction of photobleaching (Fig. 1e,f). A record without voltage pulses (−120 mV for the same duration as the experimental protocol) provided a trace with only residual photobleaching. This could be well fit by an exponential decay function with 2 or 3 terms (Clampfit or MATLAB). The fitted curve was then subtracted from all traces to completely remove residual bleaching.

The final sampling rate and the excitation duty cycle are limited by how well the digitizer and LED driver can be synchronized without external timing hardware. A limitation of this approach is the inability to filter the analog signals before acquisition; however, the signal-to-noise ratio can be improved by averaging.

## Electrophysiological data analysis

TEVC recordings were analyzed using Clampfit 11 (Molecular Devices), by plotting the immediate tail currents (recorded upon stepping to the tail voltage) against the preceding test voltages and fitting data with a Boltzmann function:

$$I_{\text{tail}}(V) = \frac{I_{\text{tail, max}} - I_{\text{tail, min}}}{1 + \exp\left(\frac{V_{0.5} - V}{k}\right)} + I_{\text{tail, min}} \tag{1}$$

where $V$ was the voltage of the test-pulse preceding the tail-pulse; $V_{0.5}$ was the half-activation potential; $I_{\text{tail,max}}$ and $I_{\text{tail,min}}$ were the limiting maximal and minimal tail currents, respectively; and $k$ was the slope.

COVG-VCF recordings were analyzed in MATLAB R2024b (MathWorks) and Excel 365 (Microsoft). Tail currents were fit to the Boltzmann distribution:

$$I_{\text{tail}}(V) = \frac{I_{\text{tail, max}} - I_{\text{tail, min}}}{1 + \exp\left(\frac{zF}{RT}\left(V_{0.5} - V\right)\right)} + I_{\text{tail, min}} \tag{2}$$

where $z$ was the effective valence, $T$ was the room temperature in K, and F and R the Faraday and Gas constants, respectively.

Bleaching was subtracted from fluorescence data in MATLAB after decimation. The $\Delta F$ was calculated by averaging the last 100 ms of the test pulse and fit to either a single ($n = 1$) or the sum of two ($n = 2$) Boltzmann distributions:

$$\Delta F(V) = \sum_{i=1}^{n} \frac{\Delta F_{\text{pos}, i} - \Delta F_{\text{neg}, i}}{1 + \exp\left(\frac{z_i F}{RT}\left(V_{0.5, i} - V\right)\right)} + \Delta F_{\text{neg}, i} \tag{3}$$

Where $\Delta F_{\text{pos}}$ and $\Delta F_{\text{neg}}$ were the $\Delta F$ asymptotes as $V$ approached plus or minus infinity, respectively.

For the sum of two Boltzmann functions, the following constraints were used while fitting:

a) $\Delta F_{\text{pos},2}$ was constrained to be equal to $\Delta F_{\text{neg},1}$
b) $V_{0.5,2}$ was constrained to be equal to the $V_{0.5}$ of channel opening from the same cell (Eq. 2)
c) $z_2$ was constrained to be equal to the $z$ of channel opening from the same cell (Eq. 2)

Fitting was performed by least squares using *Solver* in Excel.

Fractional amplitude ($f$) for component $i$ (1 or 2) was calculated as:

$$f_i = \frac{|\Delta F_{\text{neg},i} - \Delta F_{\text{pos},i}|}{|\Delta F_{\text{neg},1} - \Delta F_{\text{pos},1}| + |\Delta F_{\text{neg},2} - \Delta F_{\text{pos},2}|} \qquad (4)$$

Channel opening and closing kinetics were determined by fitting traces to the *Exponential standard* function, with one ($n = 1$) or the sum of two ($n = 2$) components in Clampfit 11 (Molecular Devices):

$$f(t) = C + \sum_{i=1}^{n} A_i \exp(-t/\tau_i) \qquad (5)$$

where $A$ was the amplitude of the exponential component, $\tau$ was the time constant, and $C$ was an offset.

$\Delta F$ kinetics were determined in MATLAB using *fminsearchbnd* (1.4.0.0, John D'Errico) using Eq. 5 with $n = 2$ and the following constraints:

a) $A_1$ was constrained to be equal to $-(C + A_2)$
b) $C$ was constrained to be equal to the $\Delta F$.

Only traces with signal-to-noise ratio above 4 were fit.

### Structure preparation and homology modeling

The human apo $K_V7.4$ structure (PDB: 7BYL)[30] was prepared by filling in the missing loop between the S3 and S4 helices of the VSD. The missing section between Y367 and D524 corresponds with the cytosolic linker between the HA and HB helices. This section is unresolved in all $K_V7$ structures, yet presents a small spatial gap in the structures; we therefore connected both ends using two glycine residues. An additional model containing the R216H mutation was created. Each $K_V7.4$ subunit was modeled in complex with calmodulin, $Ca^{2+}$ ions were inserted into each calmodulin subunit from the apo state $K_V7.1$ structure (PDB: 6UZZ)[62]. Missing loop sections were built using the modeler program[63], 200 models were created in each instance. The models were ranked based on the intersection of the *molpdf* and DOPE scores[64]. The top scoring models were then processed using the PROPKA tool to assign tautomeric states to titratable residues at a pH of 7.0[65]. The estimated $pK_a$ of H216 from the initial models ranged from 4.7 to 6.6 depending on the proximity of the histidine sidechain to the sidechain of D178 in each of the four voltage sensors. To enable potential salt-bridge interactions to D178, the residue H216 was assigned the +1-charged tautomer where both nitrogen atoms of the imidazole ring are protonated. During the simulation, as the S4 helix shifted to enable more consistent interactions between D178 and H216 (Fig. 6d), the estimated $pK_a$ of H216 in this conformation ranged from 6.9–8.0.

### Molecular dynamics simulations

The CHARMM-GUI webserver[66,67] was used to generate simulation boxes for each of the models, the box size was set to ~140 × 140 × 150 Å. In brief, the proteins were placed in a 1-palmitoyl-2-oleoyl-sn-glycero-3-phosphocholine (POPC) lipid bilayer. Water molecules with 0.15 M KCl were added, including excess $K^+$ ions to neutralize overall charge (Table S4). All systems were simulated using GROMACS 2021.4[68] with the CHARMM36 and CHARMM36m all-atom force field parameters, which are extensively validated for simulations of protein and membrane systems[69,70]. All simulations were carried out under periodic boundary conditions and no voltage difference was applied between the two membrane sides. All systems were equilibrated using the CHARMM-GUI default equilibration protocol which includes gradually withdrawing positional restraints from protein backbone and sidechain atoms. The protocol was slightly modified such that positional restraints of 100 kJ mol$^{-1}$ nm$^{-1}$ were maintained on protein sidechain and backbone atoms for an additional 5 ns, then restraints were withdrawn

from sidechain atoms but maintained on backbone atoms for a final 10 ns. Following equilibration, each system was simulated for 500 ns in triplicate with a 2 fs timestep, each replica was initiated from a unique velocity seed. Nonbonded interactions were treated using the Verlet cutoff scheme with a 1.2 nm cutoff for both van der Waals and real-space electrostatic interactions. The force-switching method was used to switch off van der Waals interactions between 10 and 12 Å. The particle mesh Ewald method[71] was used to calculate long-range electrostatic interactions. The LINCS algorithm was used to constrain hydrogen-bond lengths[72]. The pressure was maintained at 1 bar with semi-isotropic pressure coupling using the Berendsen barostat ($\tau_P = 1.0$ ps and isothermal compressibility $= 4.5 \times 10^{-5}$ bar), and the temperature was maintained at 300 K using the v-rescale thermostat ($\tau_T = 0.1$ ps)[73].

Preliminary simulation analysis revealed that the inclusion of $PIP_2$ molecules mitigated the impact of R216H mutant on S4 helix movement, this is consistent with PIP2 having a role in stabilizing the activated conformation of the VSD[74]. Therefore, we analyzed simulations that did not include $PIP_2$ molecules to better characterize the impact of the R216H mutant on S4 movement. Simulation distance and root mean squared deviation (RMSD) analyses were performed using the MDAnalysis tool[75]. Distances were measured between the Cα atoms of F146 and R216 or H216 from their respective systems. RMSD was calculated based on Cα atom positions relative to the initial structure and following superimposition. RMSD was calculated for the transmembrane segments (residues S74-Q332), here, Cα atoms of these residues were superimposed, RMSD was also calculated for the intracellular part of the S4 helix (residues 208-218) in each protein chain, here, the Cα atoms from the S1-S3 helix were superimposed. Molecular interactions involving R216 or H216 were analyzed using the ProLIF tool[76], salt-bridge interactions were calculated based on a 4.5 Å cutoff between the heavy atoms of charged groups in acidic and basic residues. Unless otherwise stated, analyses encompassed simulation frames from 200-500 ns. Mean ± SEM interaction values are obtained across the 4 voltage sensors and 3 simulation replicas ($n = 12$). Simulations were visualized using PyMOL (The PyMOL Molecular Graphics System, Version 2.5 Schrödinger, LLC).

The MD simulation checklist is provided in Table S5.

### Statistics

All experiments were replicated in oocytes from two or more different frogs (batches). Only cells that yielded both high-quality current and fluorescence traces were included in aggregates. When fluorescence and current data are compared, they were measured simultaneously from the same cells, labeled with the fluorophores. Data are reported as mean ± standard error of the mean (SEM), unless otherwise stated. Student's *t*-test (unpaired, two-tailed; Excel 365, Microsoft) was used to assess significance, and statistical significance was set at $p < 0.05$. To compare biophysical properties of $K_V7.4$ cysteine mutants with $K_V7.4$ WT, one-way ANOVA followed by a Dunnett's post hoc test was used, with $K_V7.4$ WT set as the control group (Prism v.10.0.2, GraphPad Software).

### Reporting summary

Further information on research design is available in the Nature Portfolio Reporting Summary linked to this article.

## Data availability

The source data underlying Figs. 1c, d, 2b, 3c,g, 4b, d, f, g, 5d–i, and Supplementary Figs. S1c, S4, S6, S7b, S10, S13 and S14 are provided as a Source Data file and are also available at Zenodo (https://doi.org/10.5281/zenodo.17742109). Accession codes used in this study: National Center for Biotechnology Information: NM_004700; Protein Data Bank: 7BYL[30] and 6UZZ[62]. Source data are provided with this paper.

## Code availability

*fminsearchbnd* by John D'Errico was retrieved from the MATLAB Central File Exchange (https://www.mathworks.com/matlabcentral/fileexchange/8277-fminsearchbnd-fminsearchcon). MD simulation files are available at Zenodo [https://doi.org/10.5281/zenodo.17742109].

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

## Acknowledgements

We dedicate this implementation of pulsed excitation for VCF to the memory of Ariel L. Escobar. We are grateful to H. Peter Larsson for comments on the manuscript, and members of the Elinder, Larsson, Liin and Pantazis groups for expert oocyte preparation. The clone for human K$_V$7.4 was a kind gift from Dr. Nicole Schmitt at the University of Copenhagen. The molecular dynamics simulations were enabled by resources provided by the National Academic Infrastructure for Supercomputing in Sweden (NAISS), partially funded by the Swedish Research Council through grant agreement no. 2022-06725. Funding: start-up funds from the Linköping University Wallenberg Center for Molecular Medicine / the Knut and Alice Wallenberg Foundation (A.P.); Hjärnfonden (The Swedish Brain Foundation) grants FO2024-0299 and FO2025-0364 (A.P.); Vetenskapsrådet (The Swedish Research Council) grants 2022-00574 (A.P.) and 2021-01885 (S.I.L.). This work was supported by the Italian Ministry for University and Research (MUR), Next Generation EU, Mission 4, Component 2, CUP E63C22002170007 (Project "A multiscale integrated approach to the study of the nervous system in health and disease", MNESYS) (to M.T. and F.M.).

## Author contributions

Conceptualization, S.I.L., M.T., A.P.; Methodology, A.P.; Investigation, M.N., D.J.A.F., A.S.K, K.W., S.S.Y., S.P., F.M., S.I.L., A.P.; Formal Analysis: M.N., D.J.A.F., A.S.K., K.W., F.M., S.I.L., A.P.; Writing—Original Draft, M.N., D.J.A.F., A.S.K., S.I.L., A.P.; Writing—Review & Editing, all; Funding Acquisition, S.I.L., M.T., A.P.; Resources: S.I.L., A.P.; Supervision: S.I.L., M.T., A.P.

## Funding

## Competing interests
The authors have no competing interests.
