## [Transparent Peer Review file · Nature Communications]

Two-step voltage-sensor activation of the human KV7.4 channel and effect of a deafness-associated mutation

Corresponding Author: Dr Antonios Pantazis

Version 0:

Reviewer comments:

Reviewer #1

(Remarks to the Author)

Thank you for the invitation to review: 'Two-step voltage-sensor activation of the human Kv7.4 channel and effect of a deafness-associated mutation' by Nappi and colleagues. This is a well-executed study that describes the application of voltage clamp fluorometry (VCF) to measure conformational changes of the voltage sensing domain of Kv7.4 channels, and to assess the effects of a Kv7.4 mutation (R216H within the voltage sensor) identified in patients with progressive hearing loss. The main findings/conclusions of the study are:

1. Development of a method to record VCF signals from Kv7.4 channels, along with technical improvements on previous VCF approaches in other Kv7 channels.
2. Demonstration of two components (at the kinetic, and equilibrium level) of Kv7.4 voltage sensor movements, and correlation of the slower component with the opening/activation kinetics measured from ionic currents.
3. This finding (2) drives the main conclusion that the Kv7.4 voltage sensor undergoes two sequential transitions - to an intermediate state, followed by a fully activated state - and that the second transition (intermediate to activated) is closely correlated to channel activation. This differs from other studied Kv7 channels (Kv7.2 and Kv7.3) but has similarities to features of Kv7.1.
4. The authors also demonstrate that a voltage sensor mutation (R216H) shifts the voltage dependence of both components of the fluorescence signal, to more depolarized voltages. This destabilization of the activated conformation of the voltage sensor is rationalized by molecular dynamics simulations, indicating weakened interactions between the substituted histidine, and negative charges in the S2/S3 segments.

This manuscript describes challenging experiments that are well-designed and provide useful technical advances for assessing voltage sensor dynamics in an important family of voltage-gated potassium channels.

Major comments:

1. The application of pulsed excitation for KCNQ channel VCF is a very useful development, and dramatically improved photobleaching. This will likely help considerably with many future VCF studies of channels with slow gating kinetics, or transitions like inactivation that happen over long time scales in some channel types.
2. Line 116, and subsequent interpretation: 'The presence of two kinetic components for VSD activation and deactivation indicated that the voltage sensor undergoes distinct voltage-dependent transitions...' I agree this is a possible interpretation. I wonder whether the authors can rule out the possibility of distinct populations of channels underlying different components? For example, if a significant fraction of channels do not open, or are unable to open, might this lead to a different profile for the VCF signal.
3. Interpretations about voltage sensor differences, and relationships between KCNQ channel function (eg. pages 12, 13), based on published KCNQ structures came across as overly speculative in my view. Also I was uncertain about the description of the 'resting' VSD structure in reference #29, page 13 - differences here are subtle, and I don't think there is good consensus that these structures illustrate a resting VSD conformation. I may be misunderstanding the writing here, so

please clarify.

Minor comments:

1. The manuscript is meticulously well-prepared in terms of general clarity of figures and presentation of data. These experiments can be challenging to communicate effectively, but I felt that the manuscript was easy to read and understand.
2. The discussion is quite long and perhaps too speculative relative to the extent of data provided. Also many of the figure legends are very long (the detail is welcome at this stage but could be shortened).
3. Line 177: 'R216H apparently accelerated the kinetics of channel opening, (for the depolarization to 0 mV: $\tau_{R216H}/\tau_{WT} = 0.44 \pm 0.065$, $p = 0.023$)...' I would be cautious about interpretation of this in the context of channels that potentially (likely?) have low open probability ($\tau = 1/(\alpha + \beta)$...), rather than interference by inactivation, based on the current records provided.

Reviewer #2

(Remarks to the Author)

This study investigated the gating of the K channel Kv7.4 by using electrophysiology and voltage-clamp fluorometry (VCF). The initial set of experiments identified residue N196 as the best fluorophore docking position on the S3-S4 linker to follow S4 movements. VCF measurements with AF488 at this position showed that the voltage dependence of the fluorescence signal was shifted negatively relative to the G-V curve. While the activation voltage dependence of the VCF signal amplitude was fitted with a single Boltzmann distribution, two exponentials were required to fit the kinetics. Replacing AF488 with TAMRA C6 generated VCF signals with two components in the activation voltage dependence and the kinetics. Introducing the deafness-associated mutation R216h, which is known to negatively shift the voltage dependence of Kv7.4 currents, also shifted the VCF signals in the same direction, supporting the link between the VSD movement and channel opening. Molecular Dynamics simulations indicated a higher flexibility of the lower half of the S4 in the R216H mutant. The work is presented in a clear way that is easy to follow. Of the Kv7 channels, the Kv7.1 activates in two steps via an intermediate state to the fully open state, while Kv7.2-7.3 show no intermediate state. The present study, together with previous gating current measurements by this group suggest that Kv7.4 recapitulates most closely the Kv7.1 gating mechanism.

Major points

1. Interpretation of, and information from the fluorescence experiments. The information obtained from the VCF experiments is somewhat limited. Two different fluorophores attached at position 196 of the S3-S4 linker report different signals. In addition, these fluorophores are attached via long linkers to the engineered Cys at position 196. The authors discuss this limitation in the discussion part. The conclusion is that activation occurs in two steps. Would it be possible to have more structural information from these experiments?

2. Fitting kinetics and voltage dependence.

2.1. It is obvious from looking at the fluorescence traces with TAMRA C6 that there are two kinetic components. This is however much less clear in the fluorescence traces with AF488, for both activation and de-activation, although the fitting is well documented in the regular and supplementary figures. For activation, the AF488 fluorescence traces are slightly better fitted with two exponential components than with one at the more positive potentials; for deactivation the fit is not better with two components. Regarding the voltage dependence of the fluorescence amplitude, I see the reasoning for the experiments with TAMRA, with well-separated components of different voltage dependence. For AF488 in contrast, the separation into two components is purely artificial. The $v_{1/2}$ of the F1 component is not indicated, but it must be almost identical to that of the F2 component. I wonder whether the story could be constructed without insisting on the two components of the AF488 signal.

2.2. In the text, the slow component of the TAMRA fluorescence is discussed as the component that appears at potentials more positive than -20mV and has a positive amplitude, thus it goes in the opposite direction from the fast component. Fig. S5 shows that there is at voltages -20 mV and more negative a slow component that has however a negative amplitude. As I understand, this is considered in the analysis as the component F2; I did however not find a discussion of the fact that at these voltages it has a negative amplitude. What is the possible interpretation of such a behavior, thus that the same component changes from negative to positive? Since this signal at < -20mV has a negative amplitude, does it have to be treated as a separate signal?

2.3. The fitting is shown in detail for the fluorescence traces. As information for the reader, could in one figure, also the fits be overlaid on the current traces?

Specific points

1. It should somewhere be stated that when fluorescence and current data are compared, they were measured simultaneously from the same cells, labeled with the fluorophores. I am sure this is the case, it should however be mentioned.
2. Number of experiments. Make sure that for all experiments, the number of experiments is indicated in the legends. This is the case for most, but it seems not all experiments.

3. The position of the S4 depends on the transmembrane voltage difference. In the description of the MD simulations, it should be indicated whether there was a voltage difference between the two sides of the membrane.
4. Protonation status of H216. Since an estimation of pKa values with propka had been done, the pKa value determined for H216 should be indicated in the text.
5. Lines 260-262, "Nevertheless..." it is concluded that the mutation shifts both VSD transitions as determined by VCF. Vor F2 this is not surprising, since its voltage dependence was constrained to the V0.5 value of the current. Could this passage be somewhat reformulated.
6. In Fig. 6 it is shown that in MD simulations, the lower S4 half is more flexible in the mutant than in the WT. The structural images give some ideas of movements, and in Fig. 5E a cartoon indicates that the movement of the S4 is horizontal. It would be good to determine whether the movements of the lower S4 are rather horizontal or vertical or in whatever direction, by following for example the distance to the membrane plane or to another rather fixed point of the channel.
7. It should be indicated which experiments were done with 2-electrode voltage-clamp and which with the cut-open oocyte technique.
8. Line 92, "Fig. S1a" should be "Fig. S1b"
9. The description on lines 126-129 is not sufficiently clear. Consider re-writing.
10. Lines 177-194, the passage about changes in kinetics is not clear. Are the mentioned WT data (line 178) shown in the manuscript? Instead of picking selected potentials where differences exist, it should be shown for the whole studied voltage range, where there are indeed differences introduced by the mutation. On lines 179-180, the logics are not clear. Why should inactivation prevent an effect on kinetics? Fig. 5h shows that for the ON kinetics with AF488, current and fluorescence kinetics do not match each other, whereas they do in the WT. How could this be interpreted?
11. Figure 5f, indicate statistics for the observed in V1/2 and delta-z

Version 1:

Reviewer comments:

Reviewer #1

(Remarks to the Author)

Thank you for addressing my comments. I felt the authors provided a thoughtful response and made appropriate changes to the manuscript.

Reviewer #2

(Remarks to the Author)

The authors have sufficiently addressed all my concerns.

Reviewer #1:

Thank you for your insightful suggestions and critique, which prompted us to consider additional interpretations and clarify our text. Please find our responses below, in blue italics. Major changes to the main text are highlighted yellow.

Thank you for the invitation to review: 'Two-step voltage-sensor activation of the human Kv7.4 channel and effect of a deafness-associated mutation' by Nappi and colleagues. This is a well-executed study that describes the application of voltage clamp fluorometry (VCF) to measure conformational changes of the voltage sensing domain of Kv7.4 channels, and to assess the effects of a Kv7.4 mutation (R216H within the voltage sensor) identified in patients with progressive hearing loss. The main findings/conclusions of the study are:

1. Development of a method to record VCF signals from Kv7.4 channels, along with technical improvements on previous VCF approaches in other Kv7 channels.
2. Demonstration of two components (at the kinetic, and equilibrium level) of Kv7.4 voltage sensor movements, and correlation of the slower component with the opening/activation kinetics measured from ionic currents.
3. This finding (2) drives the main conclusion that the Kv7.4 voltage sensor undergoes two sequential transitions - to an intermediate state, followed by a fully activated state - and that the second transition (intermediate to activated) is closely correlated to channel activation. This differs from other studied Kv7 channels (Kv7.2 and Kv7.3) but has similarities to features of Kv7.1.
4. The authors also demonstrate that a voltage sensor mutation (R216H) shifts the voltage dependence of both components of the fluorescence signal, to more depolarized voltages. This destabilization of the activated conformation of the voltage sensor is rationalized by molecular dynamics simulations, indicating weakened interactions between the substituted histidine, and negative charges in the S2/S3 segments. This manuscript describes challenging experiments that are well-designed and provide useful technical advances for assessing voltage sensor dynamics in an important family of voltage-gated potassium channels.

Major comments:

1. The application of pulsed excitation for KCNQ channel VCF is a very useful development, and dramatically improved photobleaching. This will likely help considerably with many future VCF studies of channels with slow gating kinetics, or transitions like inactivation that happen over long time scales in some channel types.

Thank you for appreciating both the scientific and methodological outcomes of our work.

2. Line 116, and subsequent interpretation: 'The presence of two kinetic components for VSD activation and deactivation indicated that the voltage sensor undergoes distinct voltage-dependent transitions...' I agree this is a possible interpretation. I wonder whether the authors can rule out the possibility of distinct populations of channels underlying different components? For example, if a significant fraction of channels do not open, or are unable to open, might this lead to a different profile for the VCF signal.

This is a very insightful suggestion. We have added a text to account for it in the Discussion:

"An alternative interpretation is that the two fluorescence components arise from distinct channel populations, such as conducting and non-conducting channels. For example, a structurally-locked conformation in which assembled channels are functionally dormant has been described for Kv7.3 (Zaika et al., BJ 2008), although this seems to depend on the presence of an alanine residue at position 315, which is unique for KCNQ3 and is not

conserved in KCNQ4 channels herein being studied. Thus, while we assume this is not the case here, we cannot definitively exclude this interpretation.”

3. Interpretations about voltage sensor differences, and relationships between KCNQ channel function (eg. pages 12, 13), based on published KCNQ structures came across as overly speculative in my view. Also I was uncertain about the description of the ‘resting’ VSD structure in reference #29, page 13 - differences here are subtle, and I don’t think there is good consensus that these structures illustrate a resting VSD conformation. I may be misunderstanding the writing here, so please clarify.

We apologize for our lack of clarity. To remove unnecessary speculation from our discussion, we have removed this part from the revised manuscript.

Minor comments:

1. The manuscript is meticulously well-prepared in terms of general clarity of figures and presentation of data. These experiments can be challenging to communicate effectively, but I felt that the manuscript was easy to read and understand.

Thank you for appreciating our work.

2. The discussion is quite long and perhaps too speculative relative to the extent of data provided. Also many of the figure legends are very long (the detail is welcome at this stage but could be shortened).

We agree. We have trimmed the discussion and removed speculative parts. Some figures are very busy, and we did our best to also trim the legends. In an effort to do this, we have moved all voltage dependence parameters to new table 1.

3. Line 177: ‘R216H apparently accelerated the kinetics of channel opening, (for the depolarization to 0 mV: $\tau_{R216H}/\tau_{WT} = 0.44/0.065$, $p = 0.023$)...’ I would be cautious about interpretation of this in the context of channels that potentially (likely?) have low open probability ($\tau = 1/(\alpha + \beta)$...), rather than interference by inactivation, based on the current records provided.

Thank you for pointing this out. To consider also low open probability, we referred to the work of Li and Shapiro:

“current kinetic measurements were likely affected by inactivation, observed in this mutant, and low open probability (Li & Shapiro, 2005 J Neurosci)”.

Reviewer #2

Thank you very much for your insightful review. We have addressed all your concerns, updating and adding new figures and tables. Please find our responses to your points, in blue italics, below. Major changes to the main text are highlighted yellow.

This study investigated the gating of the K channel Kv7.4 by using electrophysiology and voltage-clamp fluorometry (VCF). The initial set of experiments identified residue N196 as the best fluorophore docking position on the S3-S4 linker to follow S4 movements. VCF measurements with AF488 at this position showed that the voltage dependence of the fluorescence signal was shifted negatively relative to the G-V curve. While the activation voltage dependence of the VCF signal amplitude was fitted with a single Boltzmann distribution, two exponentials were required to fit the kinetics. Replacing AF488 with TAMRA C6 generated VCF signals with two components in the activation voltage dependence and the kinetics. Introducing the deafness-associated mutation R216h, which is known to negatively shift the voltage dependence of Kv7.4 currents, also shifted the VCF signals in the same direction, supporting the link between the VSD movement and channel opening. Molecular Dynamics simulations indicated a higher flexibility of the lower half of the S4 in the R216H mutant. The work is presented in a clear way that is easy to follow. Of the Kv7 channels, the Kv7.1 activates in two steps via an intermediate state to the fully open state, while Kv7.2-7.3 show no intermediate state. The present study, together with previous gating current measurements by this group suggest that Kv7.4 recapitulates most closely the Kv7.1 gating mechanism.

Thank you for this comment, and we would like to emphasize that the proposed gating scheme for Kv7.4 best recapitulates that for Kv7.1 complexed with KCNE1 (Barro-Soria et al., Nat Comm 2014 and Hou et al., Nat Comm 2020). By themselves, Kv7.1 and 7.4 do have distinct gating mechanisms.

Major points

1. Interpretation of, and information from the fluorescence experiments. The information obtained from the VCF experiments is somewhat limited. Two different fluorophores attached at position 196 of the S3-S4 linker report different signals. In addition, these fluorophores are attached via long linkers to the engineered Cys at position 196. The authors discuss this limitation in the discussion part. The conclusion is that activation occurs in two steps. Would it be possible to have more structural information from these experiments?

It is difficult to draw more structural information from these experiments.

In other VCF investigations, also by ourselves, the quenching mechanisms were known or engineered, and that could offer additional structural information—but this is not the case here, as in the majority of VCF studies. Yet the VCF information here, provided by (i) using multiple fluorophores, (ii) studying a clinically-relevant mutation and (iii) accessing an experimental timescale from milliseconds to seconds (the latter through a technical innovation), far exceeds that of most pioneering VCF investigations on a new channel.

In addition, we have complemented the VCF experiments with molecular dynamics simulations, providing structural rationalization for the effects of R216H. Thus our work is comparatively highly rich in both functional and structural information.

2. Fitting kinetics and voltage dependence.

2.1. It is obvious from looking at the fluorescence traces with TAMRA C6 that there are two kinetic components. This is however much less clear in the fluorescence traces with AF488, for both activation and de-activation, although the fitting is well documented in the regular and supplementary figures. For activation, the AF488 fluorescence traces are slightly better

fitted with two exponential components than with one at the more positive potentials; for deactivation the fit is not better with two components.

We cannot ignore the slow kinetic component in AF488: first, it was a prominent feature consistently observed in our experiments; second, its similarity to pore-opening kinetics, reminiscent of the $K_v7.1/E1$ signal, was especially conspicuous.

About the deactivation traces: we would argue that the double kinetic fit is evidently better—see below, for the traces shown in Fig.3e:

We should also point out, that a signal with apparently a single kinetic component does not exclude that two transitions occur with close enough kinetics.

Regarding the voltage dependence of the fluorescence amplitude, I see the reasoning for the experiments with TAMRA, with well-separated components of different voltage dependence. For AF488 in contrast, the separation into two components is purely artificial. The $v/2$ of the F_1 component is not indicated, but it must be almost identical to that of the F_2 component. I wonder whether the story could be constructed without insisting on the two components of the AF488 signal.

We do apologize for the lack of clarity—the two Boltzmann components in AF488 were actually very distinct, but their parameters were “buried” in the figure legends. We have now included all voltage-dependence parameters in a separate table (Table 1), to both “declutter” the figure legends and to make the parameters easier to track.

The $v/2$ of the F_1 component for AF488 is about -55 mV.

The $v/2$ of the F_2 component for AF488 is -13 mV.

We have also modified figures 4f & 4g to include vertical arrows pointing to the $v/2$ of each component:

For R216H, the two $v/2$ s are: F_1 : -43 mV; F_2 : 6.1 mV (see Table 1).

Since F_2 was constrained to match the voltage-dependence ($v_{1/2}$ and valence) of pore opening, we should point out that the fitting was not trivial—in other words, it is meaningful that fitting with such constraints was yet good enough.

Was it necessary or artificial? We were motivated by the parsimony of the underlying mechanism: two transitions reported in both kinetics and voltage-dependence by both fluorophores, is more plausible than three (or four) components reported exclusively by each fluorophore. In our revised manuscript, we altered the discussion (as highlighted) to better express this for a single- or double-component AF488 signal:

“The data obtained in our optical investigation with both AF488 and TAMRA C6 fluorophores revealed two movements of the KV7.4 VSD, with distinct kinetics and voltage dependence. The more parsimonious interpretation of these data is that the same two VSD transitions are reported (in terms of kinetics and voltage-dependence) by both fluorophores. An alternative premise is that the KV7.4 VSD undergoes multiple transitions: one over very negative voltages ($V_{0.5} \cong -50$ mV), producing **only** the TAMRA C6 F_1 component (Fig.4f, Table 1); another **one or two**, with **the same**, intermediate voltage-dependence ($V_{0.5} \cong -30$ mV; Fig.2b, Table 1), giving rise **only** to the **fast and slow**-AF488 signals (Fig.3a,c); and **a fourth** transition with a depolarized voltage-dependence matching that of pore opening ($V_{0.5} \cong 10$ mV), reported only as the F_2 component of TAMRA C6 (Fig.4f, Table 1). This premise requires that **three or four** distinct VSD transitions are each exclusively reported by the two fluorophores. This seems unlikely considering that the fluorophores label the same site and are similar in size.”

2.2. In the text, the slow component of the TAMRA fluorescence is discussed as the component that appears at potentials more positive than -20mV and has a positive amplitude, thus it goes in the opposite direction from the fast component. Fig. S5 shows that there is at voltages -20 mV and more negative a slow component that has however a negative amplitude. As I understand, this is considered in the analysis as the component F_2 ; I did however not find a discussion of the fact that at these voltages it has a negative amplitude. What is the possible interpretation of such a behavior, thus that the same component changes from negative to positive? Since this signal at < -20 mV has a negative amplitude, does it have to be treated as a separate signal?

This is a very insightful question. Briefly, we do not believe it is necessary to treat this as a separate signal, and figure 7c captures the most salient features from both fluorophores. To speculate a little, there may be two quenching mechanisms in the system, but we hesitate to burden the manuscript, already very information-heavy, with more speculation.

Here is our interpretation in more detail:

In the case of AF488, the two quenchers have the same effect for both transitions, resulting in overall fluorescence unquenching upon depolarization.

In the case of TAMRA, the effect of the two quenchers is different for each transition:

F_1 : both quenchers reduce fluorescence resulting in fast fluorescence decrease.

F_2 : the two quenchers have the opposite effect: quencher 1 (Q1) reduces fluorescence, while quencher 2 (Q2) increases it.

At modest depolarizations, Q1 is dominant, so F_2 is reported as slow fluorescence decrease.

At intermediate depolarizations, Q1 and Q2 influences are balanced, such that the amplitude of the slow component is zero and F_2 is essentially not reported.

At higher depolarizations, Q2 dominates, so F_2 is reported as slow fluorescence increase.

This means, the quencher in figure 7c is in fact Q2. We don't know of a simple way to make a cartoon with both quenchers, nor can we guess their identities. So while this interpretation accounts for the change in fluorescence sign, we feel it would overcomplicate the discussion.

2.3. The fitting is shown in detail for the fluorescence traces. As information for the reader, could in one figure, also the fits be overlayed on the current traces?

Yes. Fits to the current traces are now shown overlayed in figures 3b,f, S2 and S3.

Specific points

1. It should somewhere be stated that when fluorescence and current data are compared, they were measured simultaneously from the same cells, labeled with the fluorophores. I am sure this is the case, it should however be mentioned.

You are correct, this was the case. We mention this in the Methods, under Statistics:

“Only cells that yielded both high-quality current and fluorescence traces were included in aggregates. When fluorescence and current data are compared, they were measured simultaneously from the same cells, labeled with the fluorophores.”

2. Number of experiments. Make sure that for all experiments, the number of experiments is indicated in the legends. This is the case for most, but it seems not all experiments.

We have included n and batch numbers for figures 3 and S4, and in new Table 1.

3. The position of the S4 depends on the transmembrane voltage difference. In the description of the MD simulations, it should be indicated whether there was a voltage difference between the two sides of the membrane.

We did not apply a voltage difference in our simulations, to better replicate the experimental conditions under which the active VSD structure was resolved (i.e., in the absence of an electric field; Li et al., 2021, Molecular Cell 81, 25-37; PDB: 7BYL). As mentioned in the Methods (Molecular dynamics simulations):

“no voltage difference was applied between the two membrane sides”.

4. Protonation status of H216. Since an estimation of pKa values with propka had been done, the pKa value determined for H216 should be indicated in the text.

We thank the reviewer for this suggestion and have now provided pK_a estimates for H216 from the initial modelled structure and by the end of the simulation. We observed that within the top scoring models. The pK_a of H216 ranged from 4.7 to 6.6 depending on the proximity of the histidine sidechain to the sidechain of D178 in each of the four voltage sensors. As this initial conformation swaps an arginine for a histidine without accounting for resultant helical shifts, H216 exhibits conformations where it is oriented away from D178, and has a relative low pK_a as a result. During the simulation where H216 is predominantly oriented towards D178, it has a higher pK_a of 6.9-8.0. We have now amended the Methods section to include the pK_a of H216:

“The estimated pK_a of H216 from the initial models ranged from 4.7 to 6.6 depending on the proximity of the histidine sidechain to the sidechain of D178 in each of the four voltage sensors. To enable potential salt-bridge interactions to D178, the residue H216 was assigned the +1-charged tautomer where both nitrogen atoms of the imidazole ring are protonated. As

the S4 helix shifted to enable more consistent interactions between D178 and H216 (Fig.6d), the estimated pK_a of H216 in this conformation ranged from 6.9-8.0"

5. Lines 260-262, "Nevertheless..." it is concluded that the mutation shifts both VSD transitions as determined by VCF. For F2 this is not surprising, since its voltage dependence was constrained to the $V_{0.5}$ value of the current. Could this passage be somewhat reformulated.

The lines you refer-to describe the kinetic differences between fluorescence signals of AF488 between wild-type and mutant and do not explicitly comment on the shift of the transitions. We have clarified this section by removing "Nevertheless".

N.b., the main indication from VCF experiments that the mutation shifts both VSD transitions is from analysis of the TAMRA voltage-dependence, in the previous paragraph:

"we attempted to fit the ΔF from $K_v7.4^$ R216H channels constraining either F_1 or F_2 to have similar voltage-dependence parameters to those from $K_v7.4^*$ channels. This resulted in less satisfactory fits to the data (Fig.S13), supporting that R216H impairs both transitions."*

6. In Fig. 6 it is shown that in MD simulations, the lower S4 half is more flexible in the mutant than in the WT. The structural images give some ideas of movements, and in Fig. 5E a cartoon indicates that the movement of the S4 is horizontal. It would be good to determine whether the movements of the lower S4 are rather horizontal or vertical or in whatever direction, by following for example the distance to the membrane plane or to another rather fixed point of the channel.

We thank you for prompting this investigation, it has yielded quite insightful results that are now reported in new figure S16 and in the results section.

We have aligned the protein based on the S1-S3 helices positions to see how the S4 helix moves relative to them. To illustrate this, we project the coordinates of the $C\alpha$ atom of R216 along the X and Y axes to investigate horizontal motion in the plane of the membrane and in Y and Z axes to investigate vertical motion perpendicular to the membrane. This is accompanied by a projection of the $C\alpha$ atom of F143, the charge transfer center (CTC) as a reference point. We also provided 100 representative simulation frames shown in the XY and YZ plane to illustrate these observations in the structure. Projecting the coordinates on the XY plane, showed that the H216 mutant containing S4 helix exhibited more horizontal motion away from the CTC compared to the WT system. Projecting the coordinates on the YZ plane demonstrated that the movement of the of H216 away from the CTC exhibited was in the shape of an arc, i.e. the bottom of the S4 helix swung further away from the CTC in the presence of H216 mutant compared to WT. What was quite interesting here is that a small but clear population of simulation frames sampled the position of H216 below that of WT system. This may reveal that the voltage sensor has a greater tendency to go down in the presence of the mutant consistent with the right shifting effect of this mutation on the voltage of activation.

Section added to results:

"To determine whether the elevated motion of S4 containing the H216 mutant was primarily horizontal or vertical, we projected the $C\alpha$ coordinates of H216/R216 and the reference residue F143 onto the XY and YZ planes (Figure S16). In the XY projection, the H216 mutant's S4 helix showed greater horizontal displacement away from the F143 than in the WT system. The YZ projection revealed an arc-shaped motion, with the lower portion of the S4 helix swinging further from the CTC in the H216 mutant. Notably, a small but distinct subset of frames positioned H216 below its WT counterpart, suggesting that the voltage sensor more readily moves downward in the mutant, consistent with its rightward shift in activation voltage."

7. It should be indicated which experiments were done with 2-electrode voltage-clamp and which with the cut-open oocyte technique.

Thank you for prompting this clarification. In our Methods, it was already stated that all VCF experiments were performed under COVG:

“Voltage-clamp fluorometry was performed by adding epifluorescence and light detection capability to the COVG system. Prior to mounting on the COVG apparatus, oocytes were stained...”

To clarify that TEVC was used for the experiments to evaluate the voltage dependence and functional expression of Cys-substitution clones, we added under the “Two-electrode voltage clamp” section:

“The initial electrophysiological experiments to evaluate the voltage dependence and functional expression of Cys-substitution clones (Fig. 1c,d) were performed under two-electrode voltage clamp.”

We also ensured that the type of experimental technique used is mentioned in all figure legends with representative traces.

8. Line 92, "Fig. S1a" should be "Fig. S1b"

Thank you. Fixed.

9. The description on lines 126-129 is not sufficiently clear. Consider re-writing.

Thank you for urging us to clarify, we have accordingly rewritten this section:

“Whereas AF488 ΔF increased monotonically with progressive depolarization consistently as the voltage became more positive, TAMRA C6 produced a more complex signal response to depolarization (voltage increase): modest progressive depolarizations (up to about -30 mV) resulted in fluorescence decrease ($\Delta F < 0$); with stronger depolarizations, At -30 mV and above, an additional, slower fluorescence component became apparent with opposite amplitude, to the first ($\Delta F > 0$) which resulted in the ΔF increasing with progressive depolarizations (Fig. 4a-b).

10. Lines 177-194, the passage about changes in kinetics is not clear. Are the mentioned WT data (line 178) shown in the manuscript?

The ratios were not previously plotted in a figure. They were calculated from the time constants plotted in Figs. 3c and 5h. We now plotted the ratios and p-values in supplemental figure S14e,f (see also our response, below).

Instead of picking selected potentials where differences exist, it should be shown for the whole studied voltage range, where there are indeed differences introduced by the mutation.

The potentials were selected because they represent a physiologically meaningful depolarization (0 mV) and repolarization (-80 mV). These had the added benefit of having strong fluorescence signals, which could be well determined by kinetic fitting. We plotted τ_{WT}/τ_{R216H} over a voltage range in supplemental figure S14e,f.

On lines 179-180, the logics are not clear. Why should inactivation prevent an effect on kinetics?

We apologize for the lack of clarity; we did not mean that inactivation would necessarily prevent an effect on pore opening (it may or it may not). Instead, we meant a technical challenge: if inactivation kinetics were similar to those of pore opening, they would interfere with kinetic fitting, preventing accurate parameter determination. Prompted by reviewer #1, we referred also to the low open probability of these channels (Li and Shapiro, 2005 J Neurosci).

Fig. 5h shows that for the ON kinetics with AF488, current and fluorescence kinetics do not match each other, whereas they do in the WT. How could this be interpreted?

True—we believe it is due to inactivation introduced by the R216H mutation, which prevents accurate kinetic characterization (in this figure, scaling of the current with respect to the fluorescence kinetic component). We have reached the limits of what we can deduce in this study, and we feel it would be inappropriate to add more speculation here.

The apparent inactivation introduced by R216H in $K_{V7.4}$ is far from an open-and-shut case; it is in itself an interesting phenomenon that warrants future investigation.

11. Figure 5f, indicate statistics for the observed in $V_{1/2}$ and δz

Done, we have added new table S2 with this information. While not all observations were statistically significant, they all followed the same tendency of right-ward $V_{0.5}$ shifts and decreased effective valence.